# Marginal Distribution Fitting Method for Modelling Flood Extremes on a River Network

**Brian Skahill [1,*]**, **Cole Haden Smith [2] and Brook T. Russell [3]**

1   Coastal and Hydraulics Laboratory, Engineer Research and Development Center, U.S. Army Corps of Engineers, Vicksburg, MS 39180, USA
2   Risk Management Center, Institute for Water Resources, U.S. Army Corps of Engineers, Lakewood, CO 80228, USA; cole.h.smith@usace.army.mil
3   School of Mathematical and Statistical Sciences, Clemson University, Clemson, SC 29634, USA; brookr@clemson.edu
*   Correspondence: brian.e.skahill@usace.army.mil

**Abstract:** This study utilized a max-stable process (MSP) model with a dependence structure defined via a non-Euclidean distance metric, with the goal of modelling extreme flood data on a river network. The dataset was composed of mean daily discharge observations from 22 United States Geological Survey streamflow gaging stations for river basins in Missouri and Arkansas. The analysis included the application of the elastic-net penalty to automatically build spatially varying trend surfaces to model the marginal distributions. The dependence model accounted for the river distance between hydrologically connected gaging sites and the hydrologic distance, defined as the Euclidean distance between the centers of site's associated drainage areas, for all stations. Modelling the marginal distributions and spatial dependence among the extremes are two key components for spatially modelling extremes. Among the 16 covariates evaluated for marginal fitting, 7 were selected to spatially model the generalized extreme value (GEV) location parameter (for each gaging station's contributing drainage basin, its outlet elevation, centroid x coordinate, centroid elevation, area, average basin width, elevation range, and median land surface slope). The three covariates selected for the GEV scale parameter included the area, average basin width, and median land surface slope. The GEV shape parameter was assumed to be constant throughout the entire study area. Comparisons of estimates obtained from the spatial covariate model with their corresponding "at-site" estimates resulted in computed values of 0.95, 0.95, 0.94 and 0.85, 0.84, 0.90 for the coefficient of determination, Nash–Sutcliffe efficiency, and Kling–Gupta efficiency for the GEV location and scale parameters, respectively. Brown–Resnick MSP models were fit to independent multivariate events extracted from a set of common discharge data, transformed to unit Fréchet margins while considering different permutations of the non-Euclidean dependence model. Each of the fitted model's log-likelihood values indicated improved fits when using hydrologic distance rather than Euclidean distance. They also demonstrated that accounting for flow-connected dependence and anisotropy further improved model fit. In this study, the results from both parts were illustrative; however, further research with larger datasets and more heterogeneous systems is recommended.

**Keywords:** flood frequency estimation; spatial extremes; trend surfaces; variable selection; spatial dependence; river distance; hydrologic distance; extremal coefficient

## 1. Introduction

Estimating the annual exceedance probability (AEP) for extreme floods is an important problem in hydrology for dam and levee safety. In risk assessments, the probability of failure for dams and levees often depends upon the magnitude of the hydrologic loading [1]. Hence, determining credible estimates of the AEPs of extreme floods that could lead to failure is necessary. The design flood AEP for most dams and levees is $1 \times 10^{-2}$ or less frequent. In the United States (U.S.), high hazard dams are designed to pass the Probable

Maximum Flood (PMF), which typically has an AEP of $1 \times 10^{-4}$ or less frequent. In the U.S., most projects have limited flood data. The length of the observed discharge record at many sites is less than 100 years. As a result, the greatest source of error in AEP and quantile estimates for an at-site flood frequency analysis is often a limited observed discharge record. Epistemic uncertainties in the estimated AEPs for extreme floods can potentially be reduced by incorporating as much hydrologic information into the frequency analysis as reasonably possible [2–5].

Recent U.S. Army Corps of Engineers (USACE) applied research and development directed at the spatial analysis of hydrometeorological extremes has involved the development of pointwise and areal estimates of extreme precipitation [6,7] and extreme snow water equivalent (SWE) [8]. These studies have applied max-stable processes (MSPs), the stochastic process analogue of the multivariate extreme value distribution [9–13]. With their application, one can not only compute pointwise return level maps, but also model the joint distribution and more complex areal-based assessments of risk while working within the theoretically justified mathematical framework provided by extreme value theory (EVT). Areal precipitation frequency estimates derived from the application of an MSP do not require one to develop and apply empirical depth-area reduction factors to convert point to areal estimates [14]. The MSP based analyses performed by Skahill et al. [6–8] computed spatially varying pointwise estimates of extreme precipitation/SWE by leveraging gridded covariate data and employing recent advances in variable selection and model fitting [15–17].

With the application of MSPs, the extremal coefficient is a useful measure for summarizing the degree of spatial dependence among the extreme data [9,10,18]. Its values vary between one and two; a value of one indicates complete dependence, whereas a value of two corresponds to independence. It is also possible to estimate the extremal coefficient via the madogram, and these estimates are useful for MSP model checking [19]. For response variables such as precipitation, temperature, and wind speed, it is typical for the extremal coefficient to be modelled as a function of the Euclidean distance between any two locations [18,20,21].

Asadi et al. [22] introduced an MSP based model that leveraged a unique, non-Euclidean distance metric to model extreme flood data on river networks. Their approach utilized the river distance between hydrologically connected gaging sites, and the hydrologic distance, defined as the Euclidean distance between the centers of site's associated drainage areas, for all stations. The hydrologic distance accounts for shared spatially variable meteorologic events and the geomorphometry of the river basin. Each of these two distance measures can potentially differ from the Euclidean distance. Several studies have proposed alternative methods to spatially model flood frequency; however, they have either assumed independence among the extreme data or accounted for spatial dependence in a manner that does not conform with EVT, hence potentially limiting their credibility for extrapolation [23–29].

The max-stable modelling approach proposed by Asadi et al. [22] holds promise for potential enhancements to USACE's Bayesian estimation and fitting software BestFit version 1.0 [30]. At present, BestFit combines limited at-site flood data with temporal data on historic and paleofloods, spatial data from areal precipitation frequency estimates, and causal data from the application of a calibrated hydrologic model forced with rainfall frequency events. The MSP methodology they introduced could potentially yield a clearer delineation between spatial information expansion and causal information expansion data for future BestFit applications such that they better align with the original outline of the flood frequency hydrology concept [3–5]. For these reasons, we took this modelling approach in this study.

The primary contribution of this study was in demonstrating that the trend fitting methodology introduced by Love et al. [31] and applied by Skahill et al. [6–8] is also useful for modelling extreme flood data on river networks. With its application, this study was able to automatically evaluate a larger set of potential marginal modelling covariates than

were originally considered by Asadi et al. [22] in a manual manner. We also show that their treatment of extremal dependence, that involves a combination of river distance and hydrologic distance rather than Euclidean distance, resulted in a model of the extreme flood data that closely resembled what is commonly observed with dependence model summaries for MSP applications with extreme precipitation and SWE data [6–8].

## 2. Materials and Methods

### 2.1. Study Area

The 12,989-square-kilometer (km$^2$) study area consists of the drainage area upstream of the streamflow gaging stations with identifying (ID) numbers 10, 14, 18, 21, and 22, as depicted in Figure 1a. It includes parts of the Current, Little Black, Eleven Point, Spring, and Strawberry River basins and contains 22 United States Geological Survey (USGS) streamflow gaging stations (Figure 1a). Each of these five rivers are sub-basins of the Black River (Figure 1). The Black River is entirely contained within the Salem Plateau Subdivision of the Ozark Plateau Physiographic Region, which is characterized as gently rolling topography with an abundance of karst features such as springs, sinkholes, and caves [32]. Greer Spring on the Eleven Point River, Mammoth Spring on the Spring River, and Big Spring on the Current River are the three largest springs in the Ozark Plateaus whose geographic extent is depicted in Figure 1b [32].

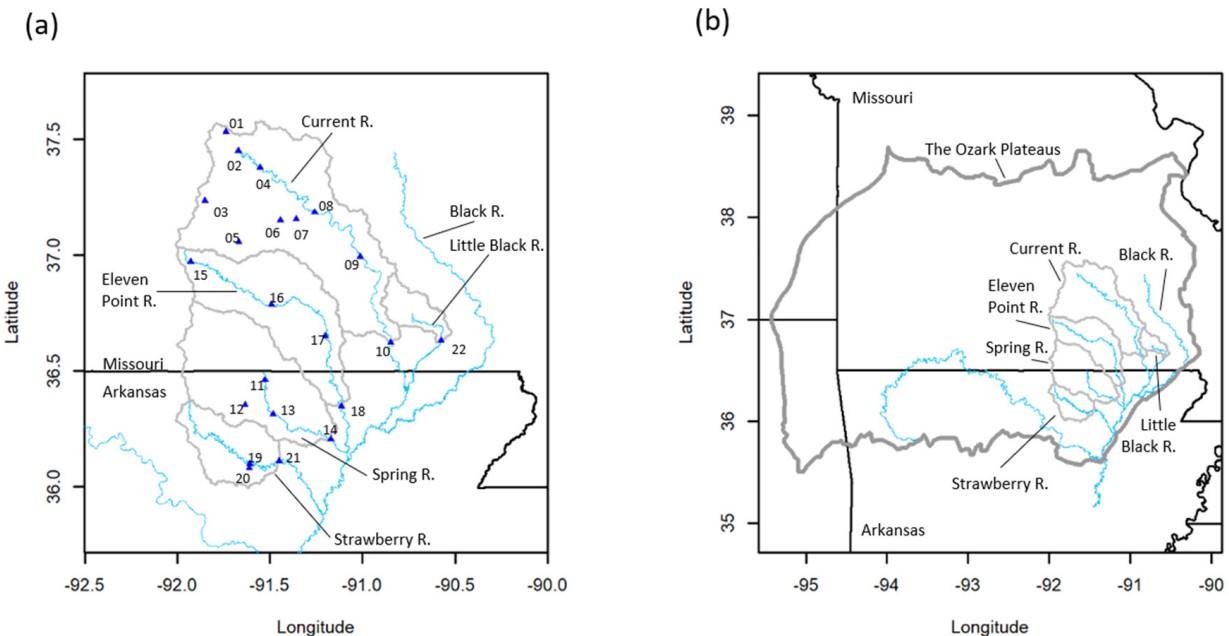

**Figure 1.** (**a**) Study area, consisting of the Current, Eleven Point, Little Black, Spring, and Strawberry River basins in Missouri and Arkansas, including the locations of 22 USGS streamflow gaging stations. (**b**) Relative location of the study area within the Ozark Plateaus. For each plot, the horizontal axis is in degrees longitude and the vertical axis is in degrees latitude.

The study area does not contain any major reservoirs, and the land area is predominantly forest and pastureland [32]. For the Current River Basin, approximately 80.1, 16, and 0.1% of its area is classified as forest, grassland, and urban land, respectively. The Little Black River Basin is classified to be approximately 54.9, 26.1, 17.7, and 0.1% forest, grassland, cropland, and urban land, respectively [33]. The Eleven Point River Basin is classified to be 65, 34, and 0.4% forest, grassland/cropland, and urban land, respectively [34]. Forest, grassland/cropland, and urban land classifications cover approximately 48.3, 49.1, and 2.4% of the Spring River Basin, respectively. The Strawberry River Basin is reported to be approximately 66 and 31.9% forest and pastureland/cropland, respectively.

Thirty-meter resolution raster datasets representative of 2001 and 2021 from the National Land Cover Database (https://www.mrlc.gov/data, accessed on 1 December 2023) were used to quantify land use changes from 2001 to 2021 for each of the five systems. Developed land increased by 0.3% within the Current River Basin and 0.2% for the remaining four basins. Forested/planted-cultivated land decreased by 0.3%/0.4%, 1.0%/0.3%, 1.6%/0.6%, 3.0%/0.8%, and 2.6%/0.8%, while shrubland/herbaceous land increased by 0.2%/0.1%, 1.4%/−0.4%, 0.8%/1.1%, 1.2%/2.4%, and 1.4%/1.8% within the Current, Little Black, Eleven Point, Spring, and Strawberry River Basins, respectively.

Adamski et al. [32] summarized the climate of the Ozark Plateaus (Figure 1b). It is characterized as temperate with its thunderstorm dominated severe weather season primarily occurring during the months from March to June. Wilkerson [33] reported the months from April to June to be the wettest for the Current and Little Black River Basins. Miller and Wilkerson [34] reported that March through May were the wettest months for the Eleven Point River Basin. Figure 2 summarizes the monthly mean precipitation climatology for each of the five basins, computed using the gridded Parameter-elevation Relationships on Independent Slopes Model (PRISM) monthly climate dataset representative of the period 1981–2010 [35]. The graphs in Figure 2 depict a trimodal distribution for the monthly mean precipitation climatology across all five basins with two larger modes occurring in May and November and a smaller mode in July. The months of April, May, and November were consistently the three wettest individual months across all five basins. Except for the Little Black River Basin, January, February, and August were consistently the three driest months. For the Little Black River Basin, January, February, June, and August were the four driest months.

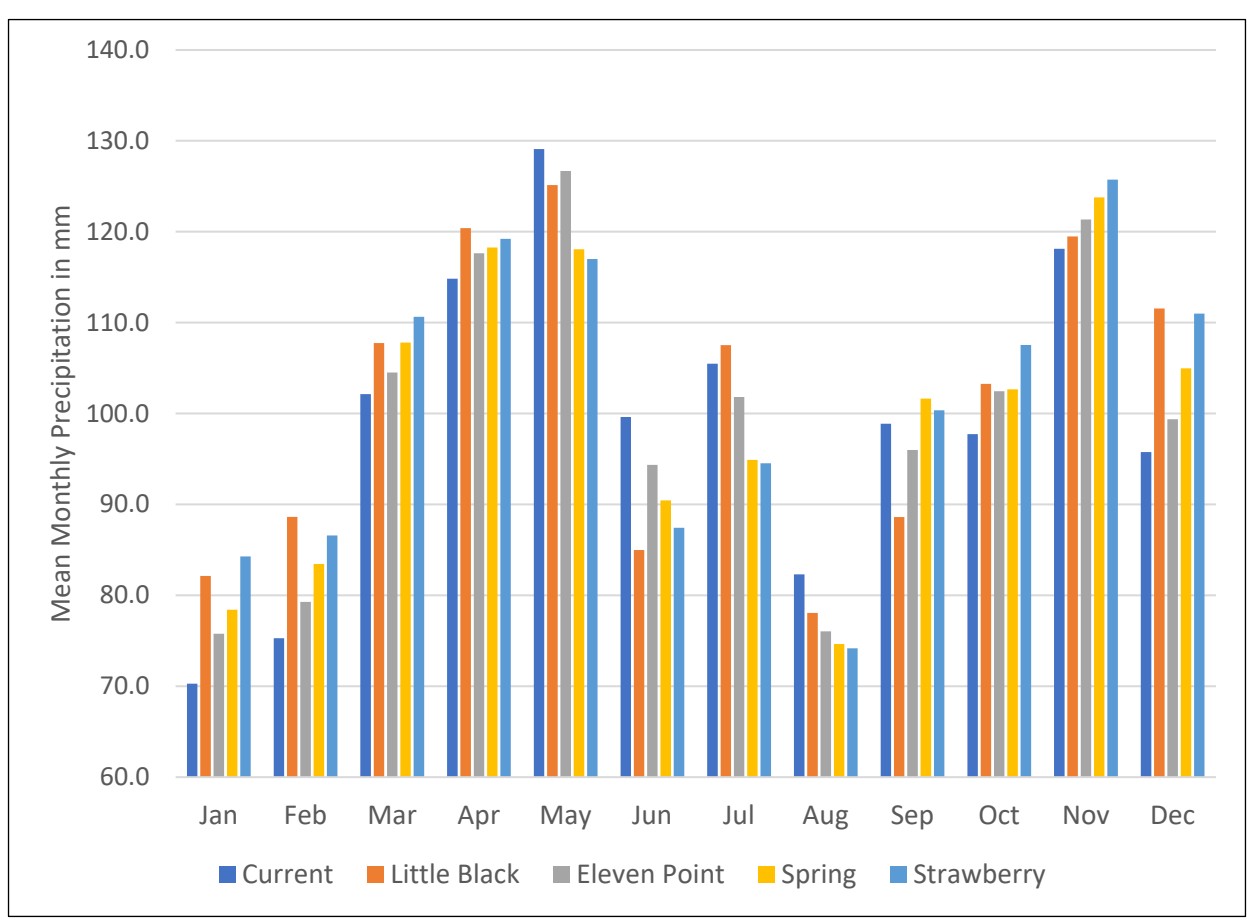

**Figure 2.** Mean monthly precipitation (units in mm) for the Current, Eleven Point, Little Black, Spring, and Strawberry River basins, computed using the gridded PRISM monthly climate dataset representative of the period 1981–2010.

Adamski et al. [32] indicated a general southeast directed increase for mean annual precipitation from minimum values in the north of the Ozark Plateaus to maximum values near its southern boundary. This trend is generally observed in Figure 3a, which depicts the mean annual precipitation computed using the PRISM monthly climate dataset. Figure 3b depicts the PRISM gridded mean annual precipitation dataset for a region surrounding the study area's five river basins. The PRISM data-computed mean annual precipitation values for the Current, Little Black, Eleven Point, Spring, and Strawberry River basins were 1189, 1217, 1195, 1199, and 1218 millimeters (mm), respectively. This large degree of homogeneity in the mean annual precipitation climatology across the five systems is observed in Figure 3.

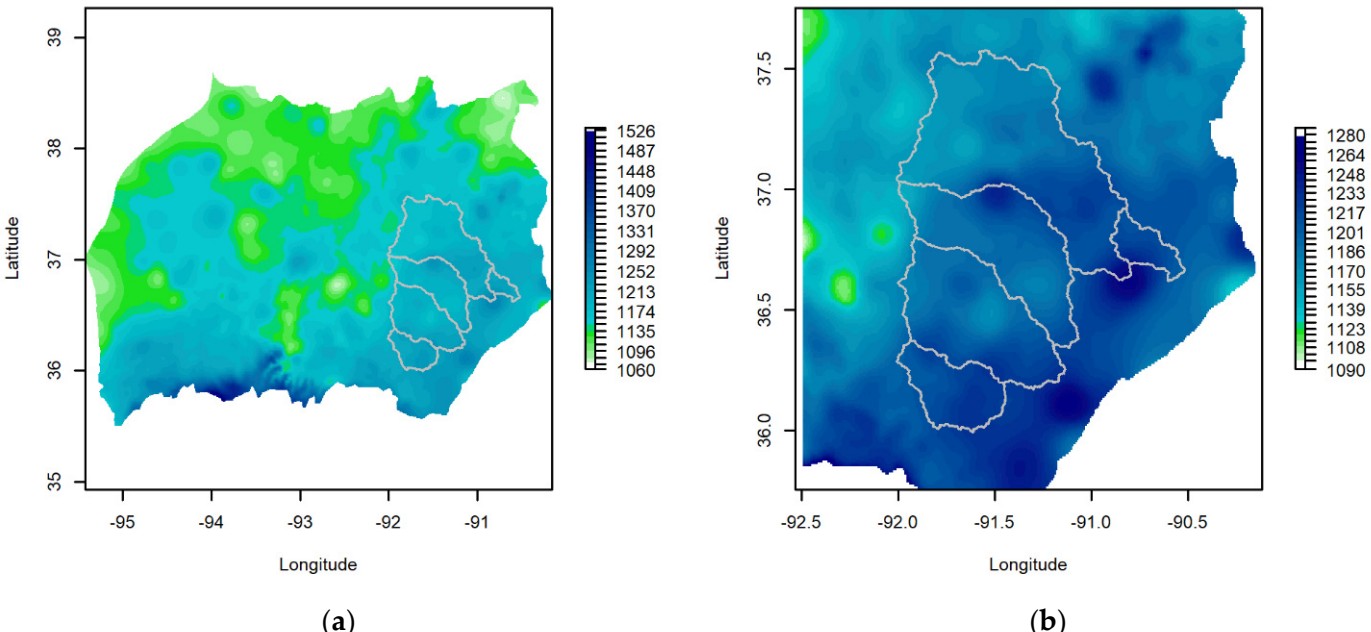

**(a)**　　　　　　　　　　　　　　　　　　　　　**(b)**

**Figure 3.** (**a**) Mean annual precipitation (values in mm) (**a**) throughout the Ozark Plateaus and (**b**) for the Current, Eleven Point, Little Black, Spring, and Strawberry River basins, delineated. The mean annual precipitation values were computed using the PRISM monthly climate data set representative of the period 1981–2010. For each plot, the horizontal axis is in degrees longitude and the vertical axis is in degrees latitude.

Adamski et al. [32] summarized mean monthly temperatures in the Ozark Plateaus to range from 30 to 38 degrees Fahrenheit (°F) during January, generally its coolest month, and from 78 to 82 °F in July, typically the warmest month. The mean monthly mean and minimum temperature values presented in Figure 4 for the Current, Little Black, Eleven Point, Spring, and Strawberry River basins, computed using the gridded PRISM dataset, support their mean temperature climatology summary [32] that January and July are the coolest and warmest months, with computed values of 0.4, 1.0, 1.0, 1.6, 2.3 and 25.2, 26.0, 25.5, 25.9, and 26.4 degrees Celsius (°C), respectively. The mean monthly mean temperature values presented in Figure 4 are above 0 °C across all months for all five watershed systems. However, the PRISM mean monthly minimum temperature values presented in Figure 4 are below 0 °C for all five river basins during the months of January, February, and December. Across the Current, Little Black, Eleven Point, Spring, and Strawberry River basins, March was consistently the fourth coolest month, with mean March minimum temperature values of 1.0, 1.9, 2.2, 1.6, and 2.7 °C, respectively.

Figures 5 and 6 and Table 1 present the spatial distribution and summary statistics of elevations and basin slopes throughout the Current, Little Black, Eleven Point, Spring, and Strawberry River Basins. The scale of the raster digital elevation model data presented in Figure 5 is one arc second, and its source is the U.S. Geological Survey 3D elevation program. The basin slopes presented in Figure 6 were computed using the digital elevation model data shown in Figure 5. For the Current, Little Black, Eleven Point, Spring, and Strawberry River Basins, the computed maximum basin reliefs were 389, 177, 374, 304, and 219 meters (m), respectively. The hypsometric curves shown in Figure 7a present a fair degree of similarity across the five watershed systems that is not easily apparent upon examination of Figure 5 and Table 1. By contrast, the plots presented in Figure 7b of basin specific land surface slopes reinforce the information presented in Figure 6 and Table 1.

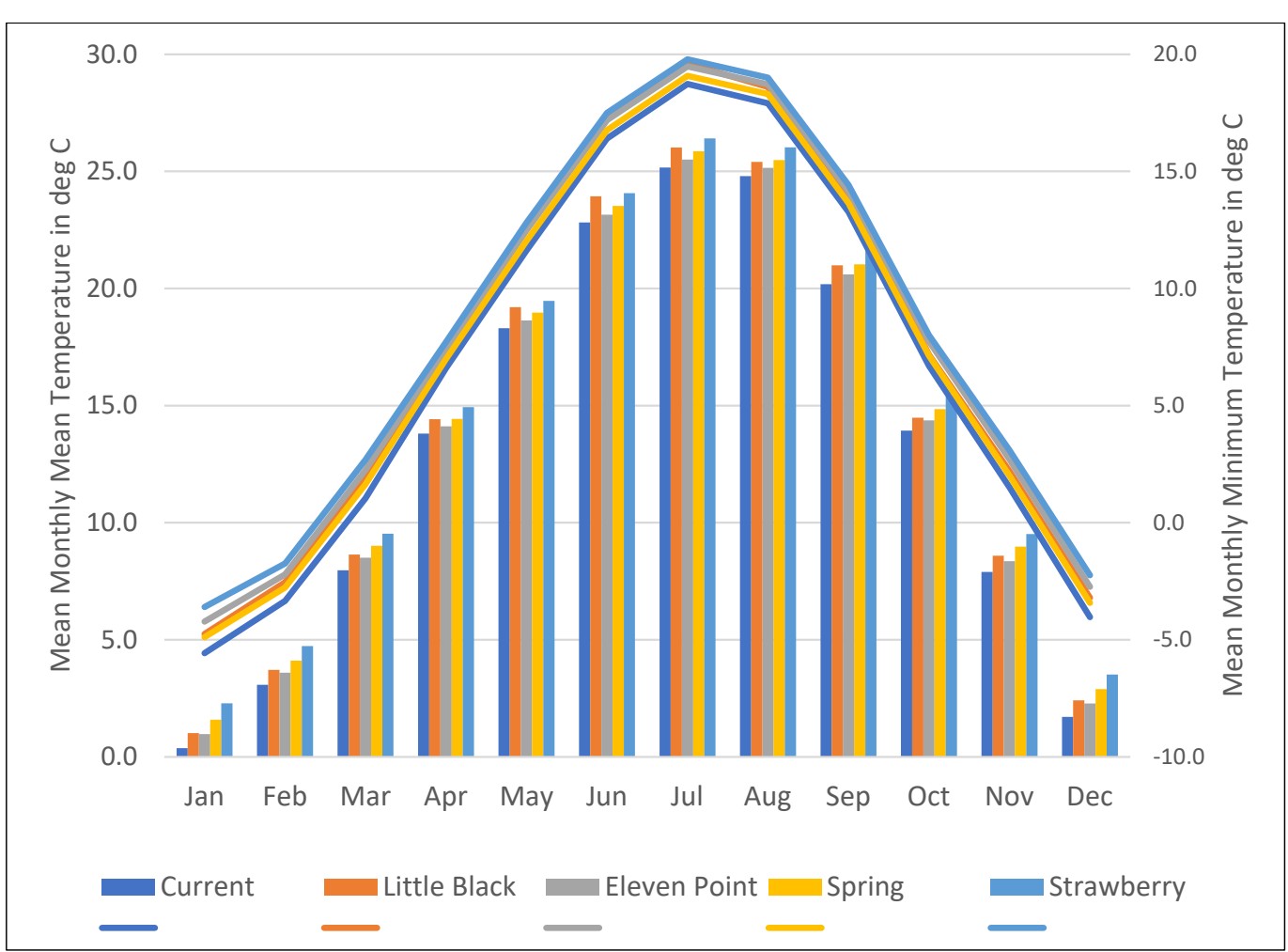

**Figure 4.** Mean monthly mean and minimum temperature (units in °C) for the Current, Eleven Point, Little Black, Spring, and Strawberry River basins, computed using the gridded PRISM monthly climate dataset representative of the period 1981–2010.

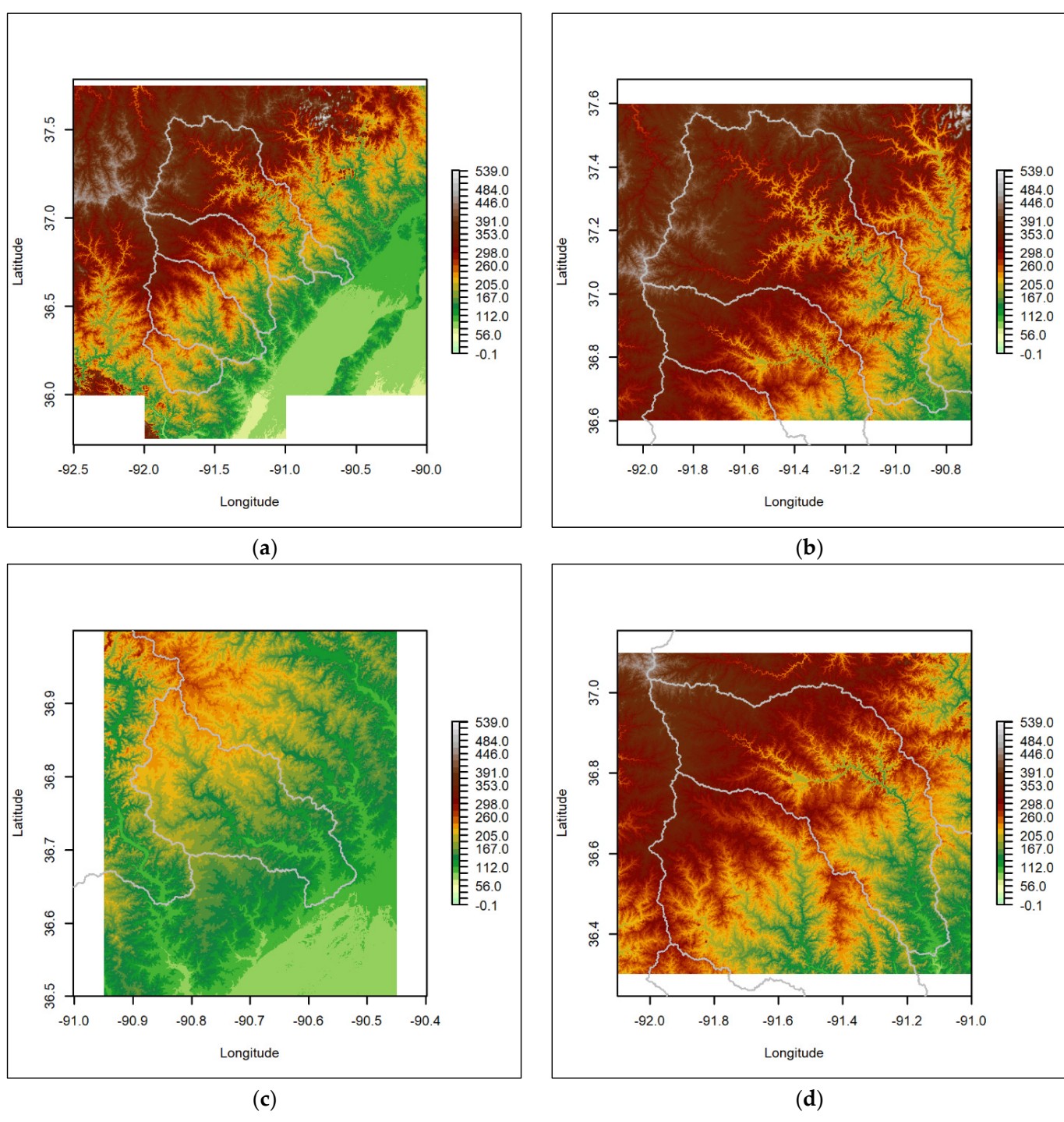

**Figure 5.** *Cont.*

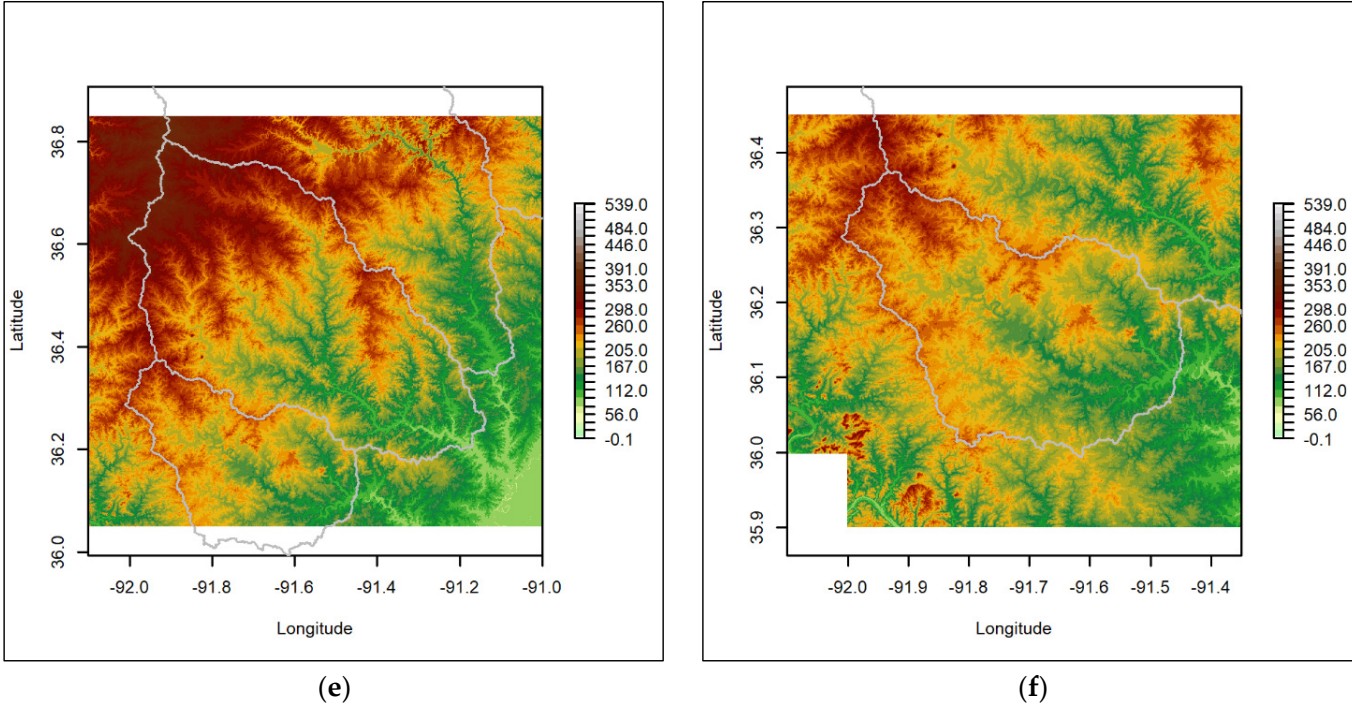

**(e)** **(f)**

**Figure 5.** Elevation values by basin (units in meters (m)). (**a**) All five modelled basins; (**b**) Current River Basin; (**c**) Little Black River Basin; (**d**) Eleven Point River Basin; (**e**) Spring River Basin; (**f**) Strawberry River Basin. One-arc-second resolution raster data from the USGS 3D elevation program was the source of the elevation values. For each plot, the horizontal axis is in degrees longitude and the vertical axis is in degrees latitude.

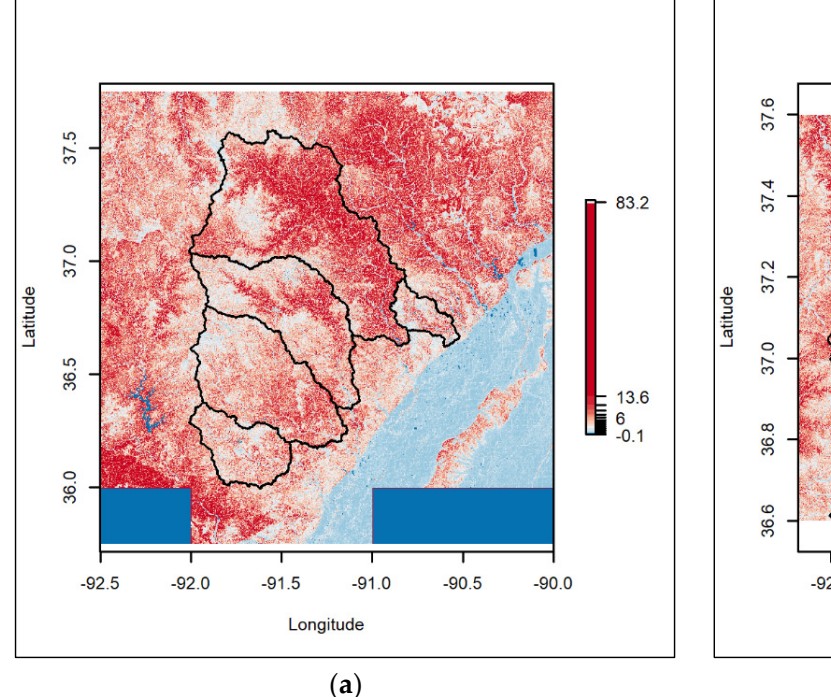 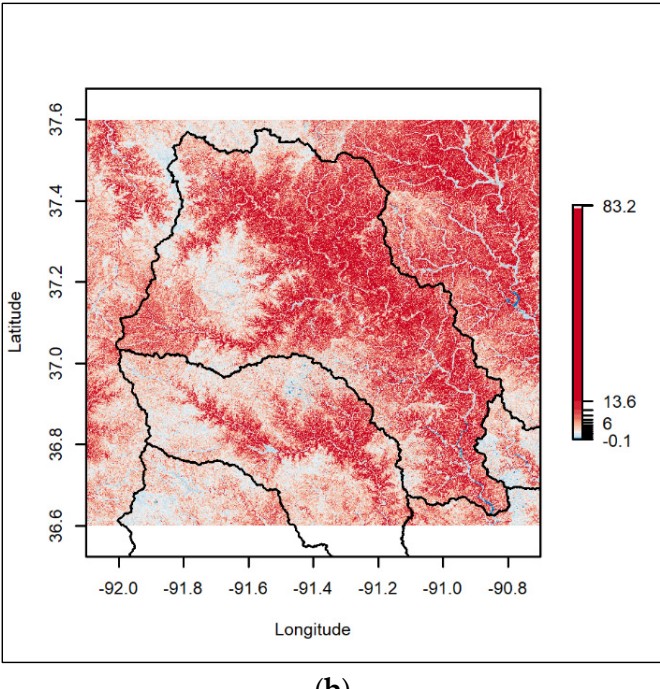

**(a)** **(b)**

**Figure 6.** *Cont.*

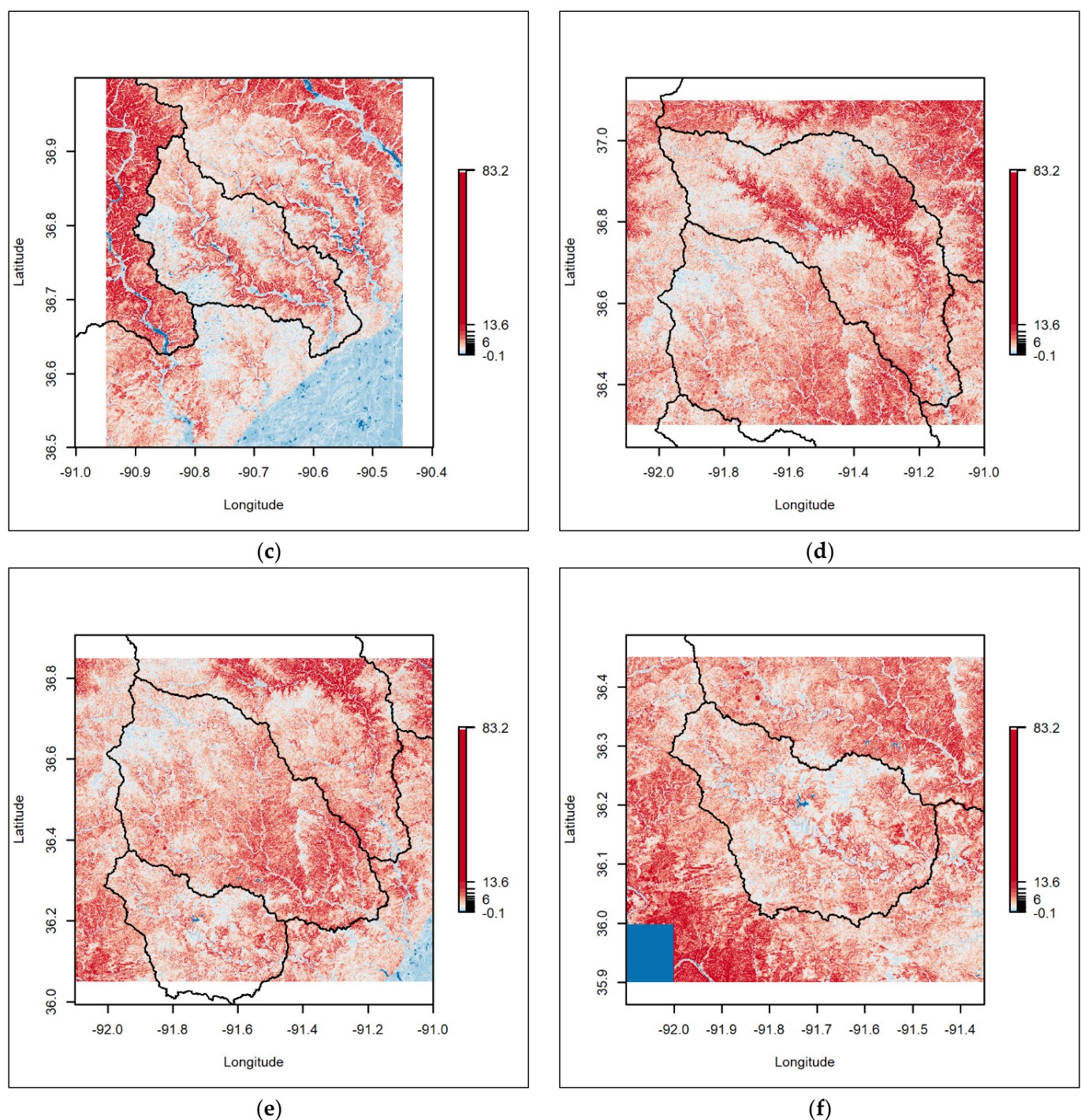

**Figure 6.** Slopes by basin (in degrees). (**a**) All five modelled basins; (**b**) Current River Basin; (**c**) Little Black River Basin; (**d**) Eleven Point River Basin; (**e**) Spring River Basin; (**f**) Strawberry River Basin. One-arc-second resolution raster data from the USGS 3D elevation program was the source of the elevation values that were used to compute the slopes. For each plot, the horizontal axis is in degrees longitude and the vertical axis is in degrees latitude.

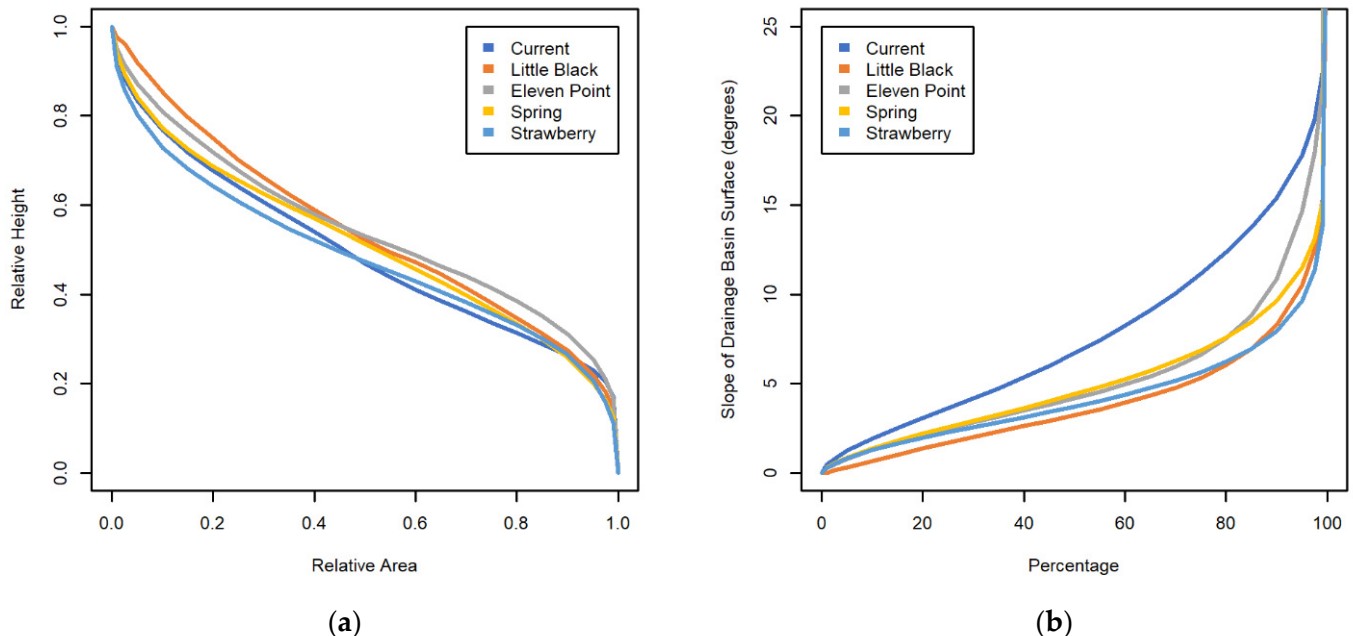

**Figure 7.** (**a**) Hypsometric curves and (**b**) surface slopes for the Current, Eleven Point, Little Black, Spring, and Strawberry River basins, computed using one-arc-second resolution raster data from the USGS 3D elevation program.

**Table 1.** Summary statistics of elevations and slopes by basin (units, meters (m) and degrees, respectively; Min. = minimum; Max. = maximum).

| | Elevations (m) | | | | | Basin Slopes (Degrees) | | | | |
|---|---|---|---|---|---|---|---|---|---|---|
| | **Min.** | **25%** | **50%** | **75%** | **Max.** | **Min.** | **25%** | **50%** | **75%** | **Max.** |
| Current | 101.80 | 241.50 | 308.00 | 359.61 | 490.82 | 0 | 3.63 | 6.70 | 11.16 | 55.18 |
| Little Black | 91.81 | 144.52 | 176.10 | 201.13 | 268.35 | 0 | 1.72 | 3.25 | 5.35 | 32.81 |
| Eleven Point | 91.15 | 212.04 | 266.58 | 310.45 | 465.59 | 0 | 2.51 | 4.19 | 6.65 | 48.03 |
| Spring | 79.25 | 183.92 | 227.03 | 271.67 | 383.35 | 0 | 2.57 | 4.42 | 6.87 | 43.04 |
| Strawberry | 90.89 | 176.54 | 205.98 | 231.41 | 309.67 | 0 | 2.29 | 3.73 | 5.67 | 38.81 |

Adamski et al. [32] summarized the streams of the Black River Basin as fast flowing with minimum and maximum monthly streamflows within the Ozark Plateaus generally occurring between July and October and between March and May, respectively. Wilkerson [33] reported flood frequencies from Alexander and Wilson [36] for several USGS streamflow gaging stations located within the Current River Basin, including for the stations with IDs 1, 3, 8, 9, and 10, as depicted in Figure 1a. They listed values of 27,300, 50,700, 68,700, 93,500, 113,000, and 185,000 cubic feet per second for 2-, 5-, 10-, 25-, 50-, and 100-year return periods for the station with ID number 10, as shown in Figure 1a. Miller and Wilkerson [34] also listed flood frequency values from Alexander and Wilson [36] for two USGS streamflow gaging stations located within the Eleven Point River Basin, viz., the stations with ID numbers 16 and 17 as depicted in Figure 1a. Southard and Veilleux [37] computed and reported flood frequency values for 14 of the 22 USGS streamflow gaging sites shown in Figure 1a, i.e., stations with ID numbers 1, 3, 6–8, 9, 10, and 15–21, as depicted in Figure 1a.

### 2.2. Discharge Data

Mean daily discharge data were collected from the USGS National Water Information System for each of the 22 USGS streamflow gaging stations whose locations are shown in Figure 1a. For each gaging station, Table 2 summarizes its assigned ID, USGS station number, location, upstream drainage area, period of record, and number of missing data

values. Figure S1 includes the period of record plots of the mean daily discharge data for each station. Based on the precipitation and temperature climatology for the study area, only the daily streamflow data from April to November were used for analysis. Table 3 lists the begin date, end date, and number of missing April–November mean daily data values for each of the 22 streamflow gaging stations. Figure S2 includes plots of the April–November mean daily discharge data for each station. Calendar year annual maxima were computed for each of the 22 streamflow gaging stations using the April–November (seasonal) mean daily data. Figure S3 includes plots of the seasonal (April–November) annual maxima for each station. Table 3 lists the number of April–November (seasonal) annual maxima for each station. Figure S4 includes plots which were used to define thresholds and decluster the seasonal mean daily discharge data to extract independent storm events for application of Equation (1). Common measurements were required for the spatial dependence modelling. Overall, 9 of the 22 stations with common April–November (seasonal) mean daily discharge measurements for the period 2002–2020 were used for the dependence modelling. Their station IDs were 04–07, 09, 10, 13, 17, and 18.

**Table 2.** Summary information for the study's 22 daily streamflow gaging stations whose locations are shown, by ID, in Figure 1 (dd = decimal degrees, # = number).

| ID | USGS Station Number | Longitude dd | Location Latitude dd | Elevation m | Upstream Drainage Area km$^2$ | Period of Record Begin Date | End Date | Missing Data # of Days |
|----|----|----|----|----|----|----|----|----|
| 01 | 7064300 | −91.73705 | 37.53023 | 354.7977 | 4.45 | 10/01/1956 | 09/30/1976 | 0 |
| 02 | 7064440 | −91.67111 | 37.44833 | 274.1449 | 152.29 | 02/07/2007 | 01/18/2021 | 1 |
| 03 | 7064500 | −91.85003 | 37.23291 | 366.1545 | 21.65 | 06/01/1949 | 10/15/1975 | 0 |
| 04 | 7064533 | −91.55281 | 37.37569 | 239.8535 | 764.05 | 08/14/2001 | 01/18/2021 | 0 |
| 05 | 7065200 | −91.66806 | 37.05611 | 257.621 | 479.15 | 10/01/2001 | 01/18/2021 | 0 |
| 06 | 7065495 | −91.44308 | 37.14817 | 200.9399 | 771.82 | 03/25/1993 | 01/18/2021 | 0 |
| 07 | 7066000 | −91.35817 | 37.15408 | 195.018 | 1030.82 | 11/01/1921 | 01/18/2021 | 0 |
| 08 | 7066500 | −91.25833 | 37.18389 | 174.3443 | 3294.47 | 08/24/1921 | 03/18/1976 | 0 |
| 09 | 7067000 | −91.01350 | 36.99139 | 136.4441 | 4317.51 | 06/18/1921 | 01/18/2021 | 0 |
| 10 | 7068000 | −90.84750 | 36.62194 | 113.4498 | 5278.40 | 06/14/1921 | 01/18/2021 | 0 |
| 11 | 7069220 | −91.52667 | 36.46028 | 137.6679 | 725.20 | 03/17/1988 | 10/04/2016 | 5665 |
| 12 | 7069295 | −91.63361 | 36.35222 | 150.2549 | 686.35 | 03/19/2010 | 01/18/2021 | 5 |
| 13 | 7069305 | −91.48278 | 36.31361 | 110.2237 | 2188.54 | 10/01/2001 | 01/18/2021 | 7 |
| 14 | 7069500 | −91.17167 | 36.20556 | 79.49311 | 3056.19 | 04/01/1936 | 01/18/2021 | 2757 |
| 15 | 7070000 | −91.92758 | 36.97039 | 358.6052 | 12.72 | 09/01/1955 | 09/30/1967 | 0 |
| 16 | 7070500 | −91.49194 | 36.78472 | 184.0703 | 934.99 | 10/01/1950 | 11/09/1976 | 0 |
| 17 | 7071500 | −91.20083 | 36.64869 | 131.1981 | 2053.86 | 10/01/1921 | 01/18/2021 | 0 |
| 18 | 7072000 | −91.11417 | 36.34639 | 93.5527 | 2926.69 | 10/01/1929 | 01/18/2021 | 2728 |
| 19 | 7073000 | −91.60833 | 36.09889 | 128.1738 | 562.03 | 03/01/1939 | 10/17/1979 | 0 |
| 20 | 7073500 | −91.61083 | 36.08056 | 129.4728 | 256.93 | 03/01/1939 | 01/30/1985 | 31 |
| 21 | 7074000 | −91.44944 | 36.11111 | 105.7302 | 1225.07 | 04/01/1936 | 09/30/2004 | 2339 |
| 22 | 7068510 | −90.57528 | 36.63167 | 91.96727 | 502.46 | 05/15/1980 | 01/18/2021 | 7464 |

**Table 3.** The begin date, end date, number of missing April–November mean daily data values and number of seasonal annual maxima for each of the study's 22 daily streamflow gaging stations whose locations are shown, by ID, in Figure 1 (# = number).

| ID | USGS Station Number | April–November Period of Record | | Missing Data # of Days | # of Seasonal Maxima |
|----|----|----|----|----|----|
| | | Begin Date | End Date | | |
| 01 | 7064300 | 04/01/1957 | 11/30/1975 | 0 | 19 |
| 02 | 7064440 | 04/01/2007 | 11/30/2020 | 0 | 14 |
| 03 | 7064500 | 04/01/1950 | 11/30/1974 | 0 | 25 |
| 04 | 7064533 | 04/01/2002 | 11/30/2020 | 0 | 19 |
| 05 | 7065200 | 04/01/2002 | 11/30/2020 | 0 | 19 |
| 06 | 7065495 | 04/01/1993 | 11/30/2020 | 0 | 28 |
| 07 | 7066000 | 04/01/1922 | 11/30/2020 | 0 | 99 |
| 08 | 7066500 | 04/01/1922 | 11/30/1975 | 0 | 54 |
| 09 | 7067000 | 04/01/1922 | 11/30/2020 | 0 | 99 |
| 10 | 7068000 | 04/01/1922 | 11/30/2020 | 0 | 99 |
| 11 | 7069220 | 04/01/1988 | 11/30/2015 | 3722 | 28 |
| 12 | 7069295 | 04/01/2010 | 11/30/2020 | 5 | 11 |
| 13 | 7069305 | 04/01/2002 | 11/30/2020 | 0 | 19 |
| 14 | 7069500 | 04/01/1936 | 11/30/2020 | 1888 | 85 |
| 15 | 7070000 | 04/01/1956 | 11/30/1966 | 0 | 11 |
| 16 | 7070500 | 04/01/1951 | 11/30/1975 | 0 | 25 |
| 17 | 7071500 | 04/01/1922 | 11/30/2020 | 0 | 99 |
| 18 | 7072000 | 04/01/1930 | 11/30/2020 | 1855 | 91 |
| 19 | 7073000 | 04/01/1939 | 11/30/1978 | 0 | 40 |
| 20 | 7073500 | 04/01/1939 | 11/30/1984 | 0 | 46 |
| 21 | 7074000 | 04/01/1936 | 11/30/2003 | 1607 | 68 |
| 22 | 7068510 | 04/01/1981 | 11/30/2020 | 4941 | 40 |

## 2.3. Covariate Data

Asadi et al. [22] evaluated four covariates to model the marginal distributions throughout $T$: the latitude of the centroid, size, mean elevation, and mean slope for the contributing drainage area associated with each gaging station. In this study, 16 covariates were evaluated for marginal fitting (Table 4). In total, 14 of the 16 covariates listed in Table 4 were readily computed for each gaging station's contributing drainage area using a geographic information system and raster digital elevation model dataset. The basin average length and average width were each computed using the estimates for the basin area and perimeter [38]. Precipitation climatology was not included as a covariate due to the large degree of homogeneity that was observed throughout the study area for the mean annual precipitation (Figure 3). Estimated values of the 16 covariates for each gaging station's contributing drainage area are provided in Table S1.

**Table 4.** The 16 covariates that were used for marginal fitting. Covariate values were readily computed for each gaging station's contributing drainage area using a geographic information system and a one-arc-second resolution raster digital elevation model dataset from the USGS 3D elevation program.

| Covariate |
|----|
| Outlet x coordinate |
| Outlet y coordinate |
| Outlet elevation |
| Centroid x coordinate |
| Centroid y coordinate |
| Centroid elevation |
| Area |
| Perimeter |
| Average length |
| Average width |

**Table 4.** *Cont.*

| Covariate |
| :---: |
| Mean elevation |
| Minimum elevation |
| Maximum elevation |
| Elevation range |
| Mean land surface slope |
| Median land surface slope |

*2.4. Methods*

In this study, we applied the EVT-based MSP modelling approach introduced by Asadi et al. [22]. The modelling analysis involved two parts, one to model the spatially variable marginal distributions and another to properly account for the spatial dependence among the observed flood data [6–8,22]. Asadi et al. [22] comprehensively outlined their MSP based modelling approach, and herein we only highlight a few of its essential features. We encourage the interested reader to refer to their work for the full details [22]. The following section, which discusses fitting the marginal distributions, includes a description of aspects that were unique to this study.

2.4.1. Marginal Fitting

In univariate EVT, it can be shown that a distribution is max-stable if and only if it is the generalized extreme value (GEV) distribution [39]. Mathematical nondegenerate limit law expressions of max-stability exist in the multivariate and spatial process settings [10]. In either case, univariate EVT results guarantee that the marginal distributions of an MSP are max-stable GEV distributions, possibly with GEV model parameters that may vary spatially. Ribatet [10,11], Ribatet et al. [9], Davison et al. [12], and Cooley et al. [13] provided thorough summaries of MSPs and MSP based modelling.

Asadi et al. [22] presented a threshold exceedance Poisson point process independence likelihood for marginal fitting:

$$L\left(\xi_j, a_{j,n}, b_{j,n}\right) \propto exp\left\{-n_j\left[1 + \xi_j\left(\frac{q_{j,p} - b_{j,n}}{a_{j,n}}\right)\right]^{-1/\xi_j}\right\} \times \prod_{i \in I_j} a_{j,n}^{-1}\left[1 + \xi_j\left(\frac{X_{j,i} - b_{j,n}}{a_{j,n}}\right)\right]^{(-1/\xi_j)-1}, \tag{1}$$

where $\xi_j, a_{j,n}, b_{j,n}, n_j$, and $q_{j,p}$ denote the GEV shape, scale, and location parameters at fixed streamflow gaging site locations on the river network, $t_j$ ($j = 1, \ldots, m$), the number of years of observations at location $t_j$, and the empirical $p$-quantile, $p \approx 1$, of the data $X_{j,i}$, $i = 1, \ldots, n$ for location $t_j$, respectively, and wherein $I_j = \{i \in \{1, \ldots, n\} : X_{j,i} > q_{j,p}\}$. With $n_j$, the parameters $\xi_j, a_{j,n}$, and $b_{j,n}$ equal those in the GEV distribution for annual maxima.

Trend surfaces were defined to support prediction throughout the entire river network, $T$. Trend surfaces spatially model the location, $\mu(s)$, scale, $\sigma(s)$, and shape, $\xi(s)$, parameters of the known GEV marginal distributions as a function of location s. For example, linear trend surfaces are of the form $\mu(s) = \eta_{\mu,0} + \eta_{\mu,1}cov_{\mu,1} + \cdots + \eta_{\mu,n_\mu}cov_{\mu,n_\mu}$, $\sigma(s) = \eta_{\sigma,0} + \eta_{\sigma,1}cov_{\sigma,1} + \cdots + \eta_{\sigma,n_\sigma}cov_{\sigma,n_\sigma}$, $\xi(s) = \eta_{\xi,0} + \eta_{\xi,1}cov_{\xi,1} + \cdots + \eta_{\xi,n_\xi}cov_{\xi,n_\xi}$, where $\eta_{\cdot,i}$ and $cov_{\cdot,i}$ are the parameters and covariates of the linear trend surface for $\mu(s)$, $\sigma(s)$, and $\xi(s)$, respectively. Factors that are assumed or known to influence extreme flood hydrology in a drainage basin, for example, climatological, morphometric, and physiographic data, were candidates to be included as covariates.

It is important to model the spatial variation of the marginal parameters by carefully "building relevant trend surfaces including any relevant covariable" [10]. Poor characterization of $\mu(s)$, $\sigma(s)$, and $\xi(s)$ complicates estimation of the dependence parameters [10,40]. In this study, linear trend surfaces for the known GEV marginal parameters were developed

by leveraging the theory of spatial extremes [9,10] and recent advances for fitting general linear models [15–17,31].

The elastic net penalty [41] was applied to regression models to facilitate model selection from among the set of potential covariate models using the trend surface fitting methodology introduced by Love et al. [31]. The elastic-net penalty is a convex combination of the penalties of ridge [42,43] and lasso [44] regression, and the resulting estimates are able to retain properties of both approaches. Given observations $y_i$, $i = 1, \ldots, n$, an $n \times m$ matrix of covariates $COV$, and an assumed linear model:

$$y_i = \eta_0 + \eta_1 cov_{i,1} + \cdots + \eta_m cov_{i,m}, \tag{2}$$

the elastic-net minimizes:

$$\frac{1}{2n}\sum_{i=1}^{n} \widetilde{w}_i \left(y_i - \eta_0 - \eta cov_i^T\right)^2 + \lambda \sum_{j=1}^{m} \left[\frac{1}{2}(1-\alpha)\eta_j^2 + \alpha|\eta_j|\right], \tag{3}$$

where $\lambda$ is non-negative and tuned to weight the penalty term; $\alpha \in [0, 1]$ controls the penalty term to vary from ridge to lasso regression at $\alpha = 0$ and $\alpha = 1$, respectively; and $\widetilde{w}_i$ is the weight assigned to the ith observation [9]. Ridge regression results in solutions that include all the predictors, whereas application of lasso regression yields sparse, much more easily interpretable solutions [45]. The elastic-net penalty is a convex combination of these two penalties. As the parameter that weights the relative contributions of the $L_1$ and $L_2$ penalties increases from 0 to 1, the number of non-zero estimated coefficients increases from 0 to the sparsity of the lasso [15].

Automatic variable selection was a primary aim for the marginal fitting analysis; therefore, we weighted the $L_1$ penalty more heavily so that the elastic-net performed much like lasso regression while retaining ridge regression's capacity to collectively shrink the coefficients for any highly correlated covariables [15,46]. To select the tuning parameter, cross validation (CV) was employed with each elastic-net model fit. Each elastic-net model was fit using the R software (version 4.2.1) package 'glmnet' [15]. For each model, the pseudo responses were made up of the three GEV univariate parameter estimates at each location, and a set of spatially varying covariates were used as covariates in the models. Independent elastic net-model fits were performed for $\mu(s)$ and $\sigma(s)$ and guided subsequent spatial GEV model fitting and selection. We note that we set $\xi(s) = \xi$, as in EVT; it is common to consider the GEV shape parameter in this manner [18,47], especially over homogeneous regions. Figure 8 is a schematic diagram depicting the main elements of the marginal fitting method.

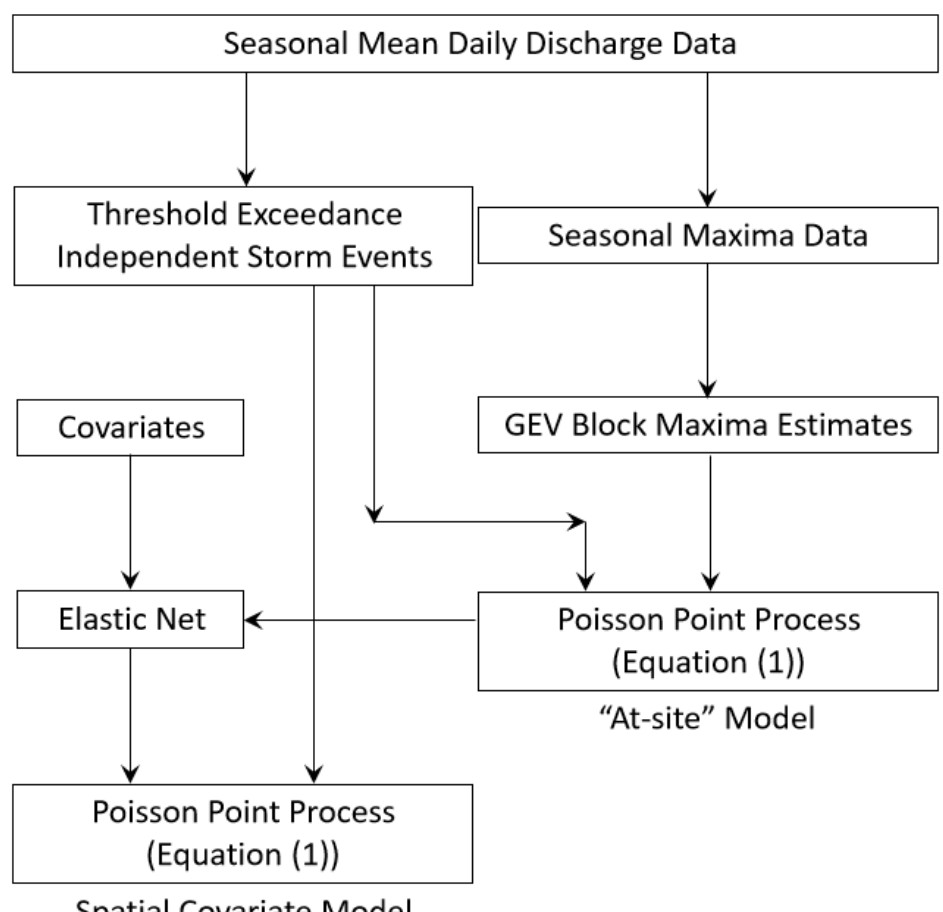

**Figure 8.** A schematic diagram that presents the main elements of the marginal distribution fitting method.

### 2.4.2. Dependence Model

Asadi et al. [22] introduced an MSP dependence model for extreme flood data. Their modelling approach aimed to account for both the river distance between hydrologically connected gaging stations and for stations that are not connected but share common meteorological events. The former is simply the distance along the river, whereas the latter is termed the hydrologic distance and is defined to be the Euclidean distance between the weighted (e.g., using precipitation climatology or elevation) centroids of their upstream drainage areas. The overall distance metric that combines both river distance and hydrologic distance is defined as:

$$
\begin{aligned}
\Gamma(s,t) &= \lambda_{Riv}\Gamma_{Riv}(s,t) + \lambda_{Hydro}\Gamma_{Hydro}(s,t) \\
&= \lambda_{Riv}\left\{1 - \prod \sqrt{\pi_{s,t}}\left(1 - \frac{d(s,t)}{\tau}\right)_{+}\right\} + \lambda_{Hydro}\|R \cdot H(s) - R \cdot H(t)\|_{2}^{\alpha}
\end{aligned}
\tag{4}
$$

for any $s,t \in T$ where $\lambda_{Riv} \geq 0$, $\lambda_{Hydro} \geq 0$, $\pi_{s,t}$, $d(s,t)$, $\tau > 0$, $R = R(\beta,c)$, $H$, and $\alpha \in (0, 2]$ represent a weight that is assigned to the dependence term for flow-connected gaging stations ($\Gamma_{Riv}(s,t)$), a weight assigned to the dependence term for gaging sites that are not flow-connected ($\Gamma_{Hydro}(s,t)$), weights that account for the proportions of extreme flood discharge values coming from each branch of the river network, the river distance between sites $s,t \in T$, the distance beyond which inter-site correlation is essentially zero, a rotation and dilation matrix to account for geometric anisotropy, the hydrological location of a gaging location on the river network, and a variogram shape parameter, respectively. Understanding the desire to model any location in $T$, observed or not, Asadi et al. [22] suggested the use of elevation as a surrogate for precipitation and that values for $\pi$ be

estimated by integrating elevations for the area upstream of each gaging station. Similarly, they suggested the hydrologic location be defined as the center of mass of the precipitation climatology (or elevation, as a replacement for precipitation) for each gaging site's contributing drainage area [22]. In order to account for potential anisotropy, the rotation and dilation matrix $R$ is given by:

$$R = \begin{pmatrix} \cos\beta & -\sin\beta \\ c\sin\beta & c\cos\beta \end{pmatrix}, \ \beta \in \left[\frac{\pi}{4}, \frac{3\pi}{4}\right], c > 0. \tag{5}$$

The parameters $\lambda_{Riv}$, $\lambda_{Hydro}$, $\tau$, $\beta$, $c$, and $\alpha$ were estimated via the fitting of the MSP. Large values of $\Gamma(s,t)$ corresponded to weak dependence, whereas small values correspond to strong dependence.

The dependence measure $\Gamma_{Riv}(s,t)$ was constructed in the following manner. We defined $\Gamma_{Riv}(s,t) = 1 - \prod \sqrt{\pi_{s,t}}\left(1 - \frac{d(s,t)}{\tau}\right)_{+}$ if $s$ and $t$ were flow-connected and $\Gamma_{Riv}(s,t) = 1$ otherwise [22]. The weights $\sqrt{\pi_{s,t}}$ reflect the number of bifurcations that occur in the river network between the two locations. We used the "linear with sill" covariance function given by $\left(1 - \frac{d(s,t)}{\tau}\right)_{+}$ [22]. For additional background on this and other covariances on river networks, refer to Ver Hoef et al. [48] and Ver Hoef and Peterson [49].

## 3. Results

### 3.1. Marginal Fitting

Using the independent storm events that were extracted from the seasonal mean daily discharge data, the Poisson point process likelihood of Equation (1) was applied to compute unique estimates for the marginal distribution's GEV parameter values at each of the 22 streamflow gaging stations. Initial estimates for the application of Equation (1) were obtained from the results of GEV at-site block maxima analyses that used the seasonal maxima data shown in Figure S3. The results obtained from applying Equation (1) without covariates are listed in Table 5.

**Table 5.** GEV model parameter estimates obtained from application of Equation (1), using the independent storm events that were extracted from the seasonal mean daily discharge data, for each of the study's 22 streamflow gaging station sites whose locations are shown, by ID, in Figure 1. Each set of GEV parameter estimates is only applicable at its respective gaging site.

| | | GEV | | |
|---|---|---|---|---|
| ID | USGS Station Number | Location | Scale | Shape |
| 1 | 7064300 | 7.83070 | 6.593739 | 0.203143061 |
| 2 | 7064440 | 651.81687 | 464.175609 | 0.344217689 |
| 3 | 7064500 | 201.25654 | 228.860738 | −0.144267974 |
| 4 | 7064533 | 2220.88580 | 2732.279987 | 0.233906106 |
| 5 | 7065200 | 3502.68845 | 4008.151359 | −0.195715630 |
| 6 | 7065495 | 3514.90339 | 3934.048180 | 0.296018442 |
| 7 | 7066000 | 3685.56258 | 3657.132441 | 0.246881711 |
| 8 | 7066500 | 8518.98331 | 9928.801022 | −0.002357151 |
| 9 | 7067000 | 10,000.00034 | 13,184.621414 | 0.107549044 |
| 10 | 7068000 | 12,469.20565 | 12,111.606256 | 0.208479773 |
| 11 | 7069220 | 365.82757 | 2372.683454 | −0.018406653 |
| 12 | 7069295 | 2805.65199 | 1452.731942 | 0.669369529 |
| 13 | 7069305 | 6590.52164 | 4115.879261 | 0.753694210 |
| 14 | 7069500 | 8379.45515 | 8044.792143 | 0.308929673 |
| 15 | 7070000 | 51.56659 | 74.159390 | −1.150904107 |
| 16 | 7070500 | 1096.32359 | 2370.342979 | −0.243435530 |
| 17 | 7071500 | 3217.66916 | 3532.282701 | 0.359971090 |
| 18 | 7072000 | 4238.21201 | 5571.942438 | 0.244704406 |
| 19 | 7073000 | 1488.63406 | 6172.408812 | −0.460305499 |

**Table 5.** *Cont.*

| | | GEV | | |
|---|---|---|---|---|
| **ID** | **USGS Station Number** | **Location** | **Scale** | **Shape** |
| 20 | 7073500 | 1307.22777 | 1423.849136 | −0.059088878 |
| 21 | 7074000 | 4722.16930 | 5034.295129 | 0.049622116 |
| 22 | 7068510 | 1566.40636 | 1946.957383 | 0.208274893 |

Figure 9 summarizes the application of two independent elastic-net regression models that were used to identify trend surface covariates for the GEV location and scale parameters. Each model used the data listed in Table 5 and the set of covariate values (standardized) listed in Table S1, and weights were assigned in accordance with the number of exceedances at each station.

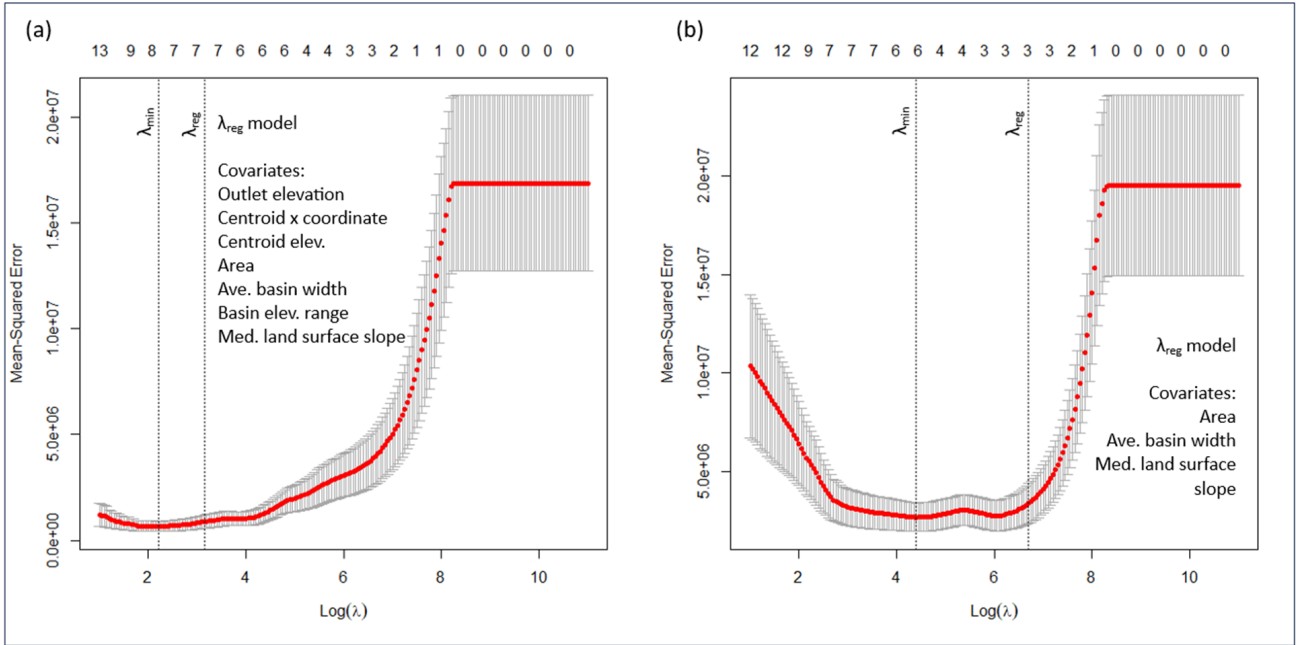

**Figure 9.** Elastic-net cross-validation (CV) plots for the study area that summarize the results for (**a**) $\mu(\text{cov}\mu)$) and (**b**) $\sigma(\text{cov}\sigma)$), identifying trend surface covariates for the GEV location and scale parameters. Each model used the data listed in Table 5, the set of covariate values (standardized) listed in Table S1, and weights which were assigned in accordance with the number of exceedances at each station. The elastic-net CV simulations considerd 16 total covariables. The *x*-axis is the natural logarithm of $\lambda$, the *y*-axis is the mean squared error (MSE), the top of the plot indicates the number of non-zero covariates as $\lambda$ varies, the red markers are the CV-derived MSE with error bars indicating one standard error, and the dotted vertical lines indicate the locations of the CV-identified $\lambda$-value that minimizes the MSE ($\lambda_{\text{min}}$) and identifies the defined most regularized model ($\lambda_{\text{reg}}$) [15].

Covariate coefficient estimates obtained from the elastic-net regression models were used as initial values for a second optimization of Equation (1). The GEV location and scale parameters were allowed to spatially vary as a function of their covariate values, and the GEV shape parameter was specified to be constant throughout the river network. Figure 10 plots comparisons of estimates obtained from the spatial covariate model with their corresponding "at-site" estimates (Table 5) for the GEV location and scale parameters at the 22 streamflow gaging stations. Figure 11 presents probability plots obtained from application of Equation (1), without and with covariates, respectively, for each of the study's 22 daily streamflow gaging stations (Figure 1, Table 2).

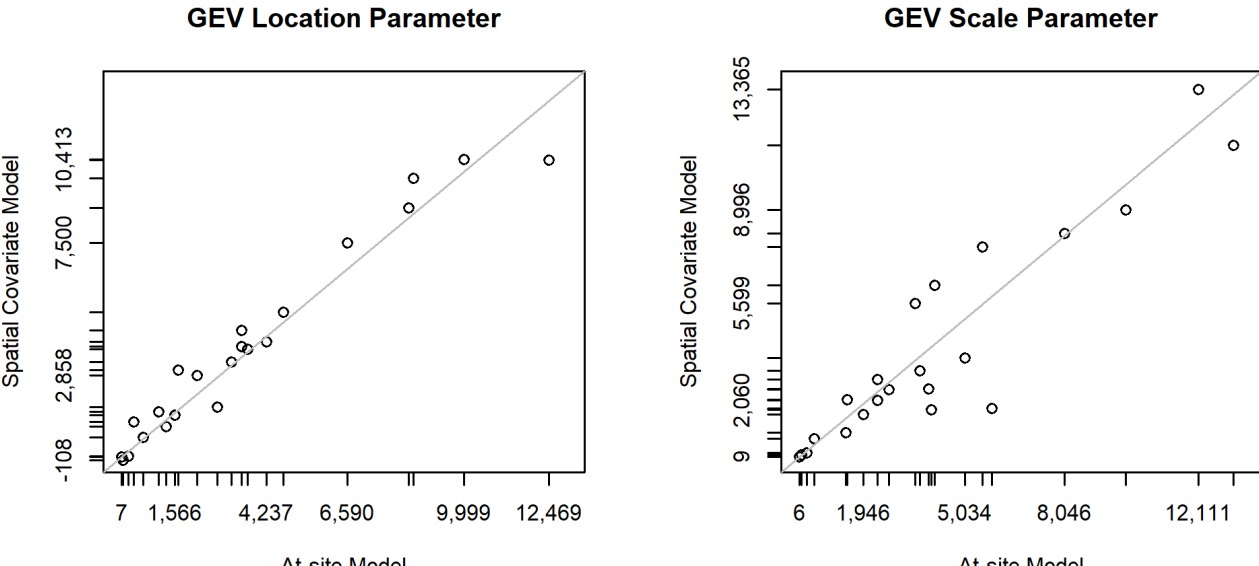

**Figure 10.** Comparisons of the trend surface models for the GEV location and scale parameters, using the spatial covariates identified from application of the elastic-net penalty, with their corresponding at-site estimates at the study's 22 streamflow gaging stations whose locations are shown, by ID, in Figure 1. In each case, results were obtained from application of Equation (1), with and without covariates, respectively.

Tables 6 and 7 summarize the results from 500 CV supervised elastic net optimization runs that were performed, in each case, for the GEV location, scale, and shape parameter. For each GEV distribution parameter, 100 CV supervised elastic net optimization runs were performed while the number of folds were set equal to 3, 9, 11, 15, and 22. Leave-one-out CV equated with the case when the number of folds equaled 22. Given the marginal trend fitting method objective was feature selection, $\alpha$ (Equation (3)) remained fixed and close in value to one for each CV-directed elastic net run, while the number of folds was allowed to vary to examine the procedure's sensitivity for covariate selection.

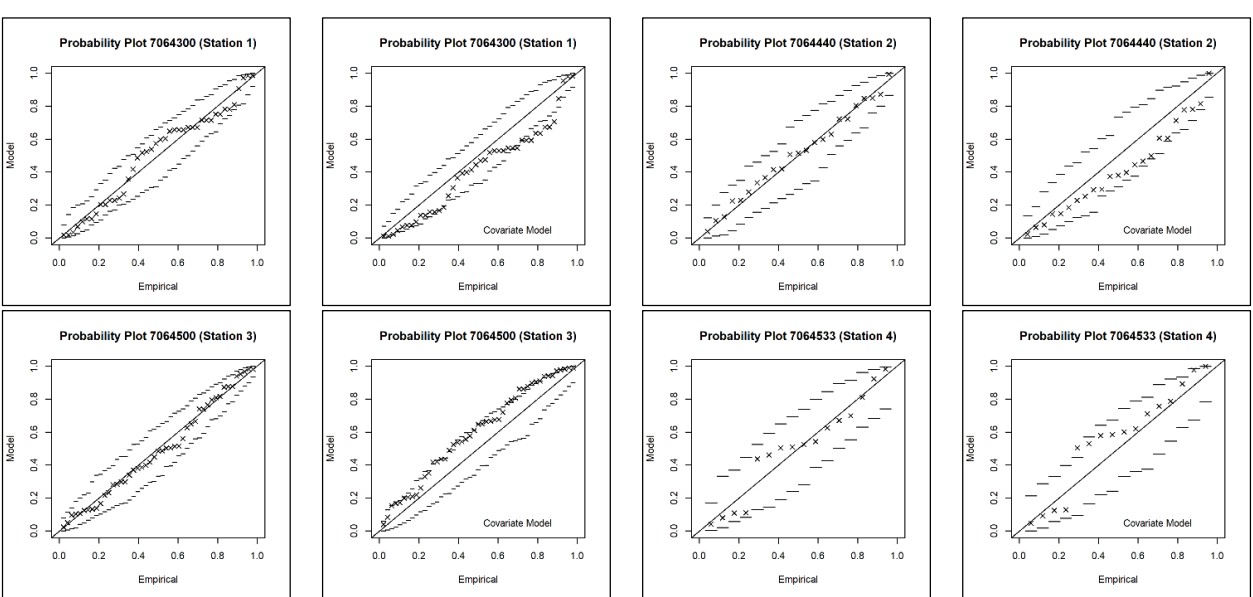

**Figure 11.** *Cont*.

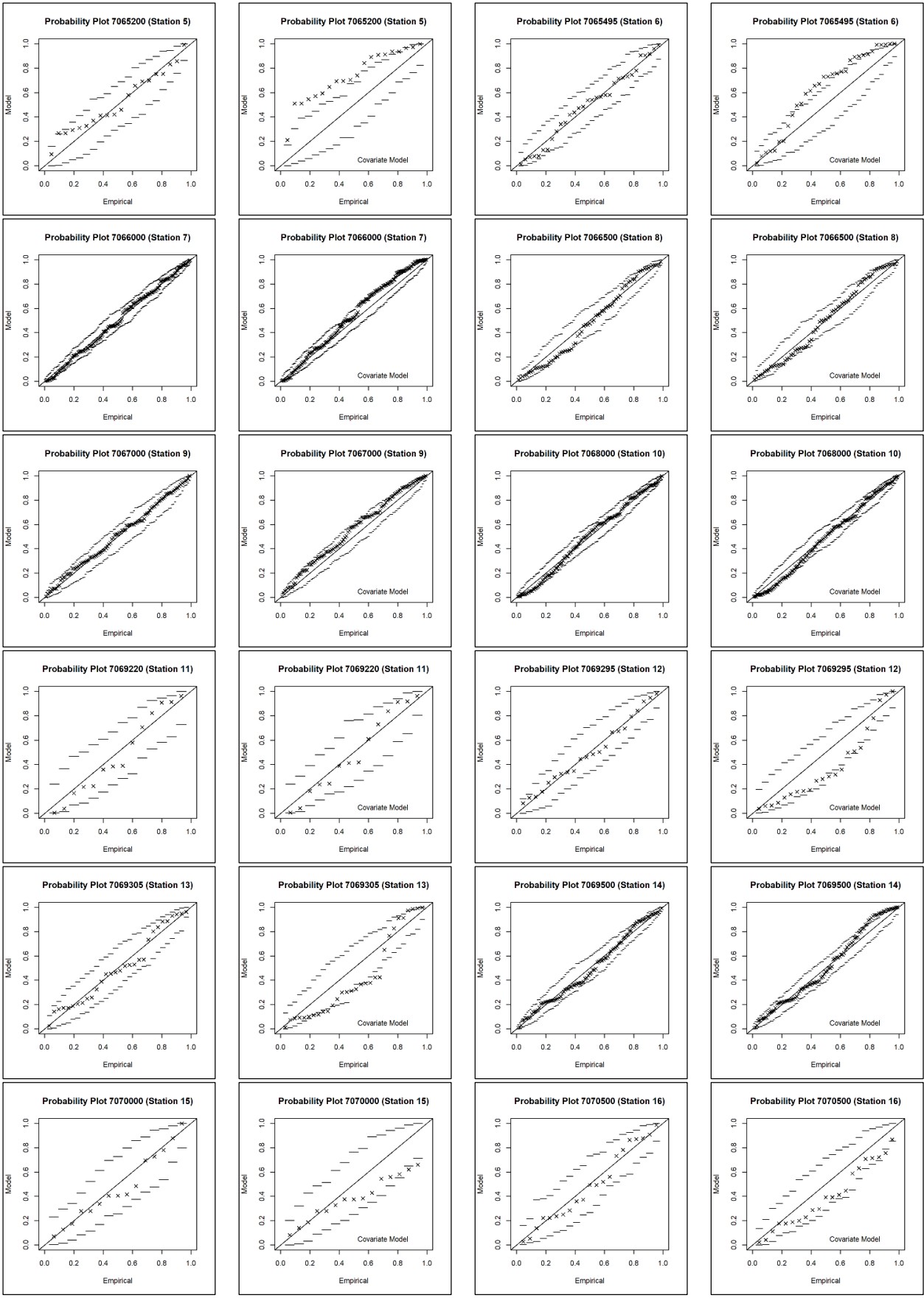

**Figure 11.** *Cont.*

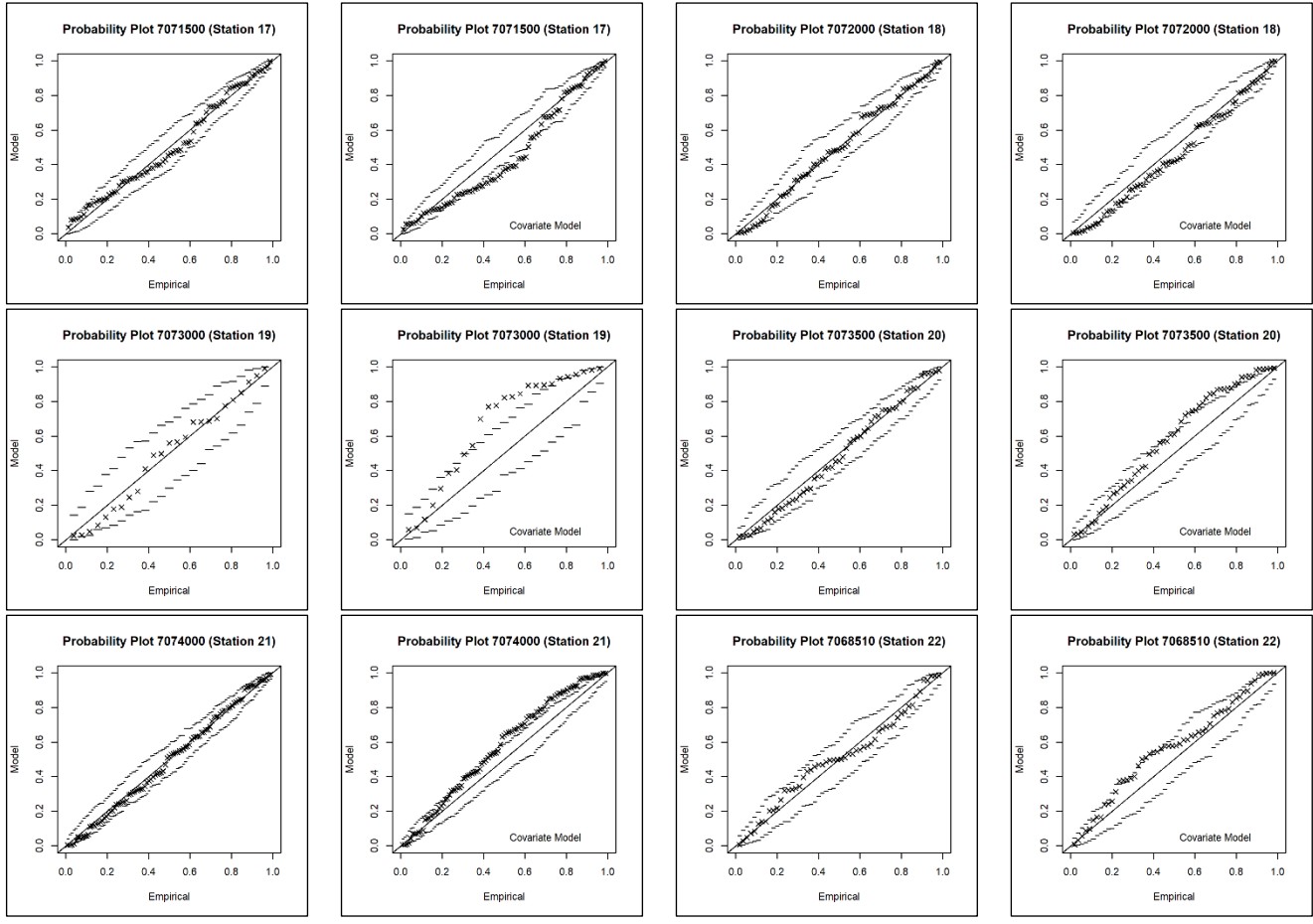

**Figure 11.** Probability plots for each of the study's 22 daily streamflow gaging stations whose locations are shown, by ID, in Figure 1. For each site, results were obtained from application of Equation (1), without and with covariates, respectively (− = 95% confidence intervals).

**Table 6.** A summary of 1500 nfold cross validation directed elastic net optimization runs with $\alpha$ fixed close in value to one (Equation (3)). For each GEV distribution parameter (location, scale, shape) and fold value (3, 9, 11, 15, 22), 100 runs were performed. Leave-one-out cross validation equates with the case when the number of folds equaled 22. The table summarizes the count associated with each covariate for each set of 100 runs.

| | GEV Location | | | | | GEV Scale | | | | | GEV Shape | | | | |
|---|---|---|---|---|---|---|---|---|---|---|---|---|---|---|---|
| | **Number of Folds** | | | | | | | | | | | | | | |
| **Covariate** | **3** | **9** | **11** | **15** | **22** | **3** | **9** | **11** | **15** | **22** | **3** | **9** | **11** | **15** | **22** |
| Intercept | 100 | 100 | 100 | 100 | 100 | 100 | 100 | 100 | 100 | 100 | 100 | 100 | 100 | 100 | 100 |
| Outlet x coordinate | 0 | 0 | 0 | 0 | 0 | 1 | 0 | 0 | 0 | 0 | 6 | 23 | 13 | 12 | 0 |
| Outlet y coordinate | 3 | 0 | 0 | 0 | 0 | 1 | 0 | 0 | 0 | 0 | 7 | 5 | 3 | 3 | 0 |
| Outlet elevation | 34 | 60 | 66 | 77 | 100 | 1 | 0 | 0 | 0 | 0 | 0 | 0 | 0 | 0 | 0 |
| Centroid x coordinate | 76 | 99 | 100 | 100 | 100 | 7 | 0 | 0 | 0 | 0 | 10 | 13 | 19 | 14 | 0 |
| Centroid y coordinate | 0 | 0 | 0 | 0 | 0 | 2 | 0 | 0 | 0 | 0 | 9 | 5 | 3 | 3 | 0 |
| Centroid elevation | 96 | 100 | 100 | 100 | 100 | 0 | 0 | 0 | 0 | 0 | 1 | 0 | 1 | 1 | 0 |
| Area | 100 | 100 | 100 | 100 | 100 | 100 | 100 | 100 | 100 | 100 | 2 | 0 | 1 | 1 | 0 |
| Perimeter | 6 | 0 | 0 | 0 | 0 | 0 | 0 | 0 | 0 | 0 | 0 | 0 | 0 | 0 | 0 |
| Average length | 2 | 0 | 0 | 0 | 0 | 0 | 0 | 0 | 0 | 0 | 0 | 0 | 0 | 0 | 0 |
| Average width | 100 | 100 | 100 | 100 | 100 | 100 | 100 | 100 | 100 | 100 | 1 | 0 | 0 | 0 | 0 |
| Mean elevation | 0 | 0 | 0 | 0 | 0 | 1 | 0 | 0 | 0 | 0 | 1 | 0 | 0 | 0 | 0 |
| Minimum elevation | 0 | 0 | 0 | 0 | 0 | 0 | 0 | 0 | 0 | 0 | 38 | 45 | 41 | 34 | 0 |
| Maximum elevation | 5 | 0 | 0 | 0 | 0 | 28 | 5 | 4 | 5 | 0 | 0 | 0 | 0 | 0 | 0 |
| Elevation range | 69 | 99 | 100 | 100 | 100 | 8 | 0 | 0 | 0 | 0 | 41 | 45 | 41 | 34 | 0 |

**Table 6.** *Cont.*

| Covariate | GEV Location | | | | | GEV Scale | | | | | GEV Shape | | | | |
|---|---|---|---|---|---|---|---|---|---|---|---|---|---|---|---|
| | Number of Folds | | | | | | | | | | | | | | |
| | 3 | 9 | 11 | 15 | 22 | 3 | 9 | 11 | 15 | 22 | 3 | 9 | 11 | 15 | 22 |
| Mean land surface slope | 0 | 0 | 0 | 0 | 0 | 2 | 0 | 0 | 0 | 0 | 3 | 0 | 1 | 1 | 0 |
| Median land surface slope | 100 | 100 | 100 | 100 | 100 | 91 | 100 | 100 | 100 | 100 | 5 | 5 | 2 | 2 | 0 |

**Table 7.** Summary statistics associated with 1500 nfold cross validation directed elastic net optimization runs with $\alpha$ fixed close in value to one (Equation (3); Table 6). For each GEV distribution parameter (location, scale, shape) and nfold value (3, 9, 11, 15, 22), 100 runs were performed. Leave-one-out cross validation equated with the case when the number of folds equaled 22. The summary statistics were computed using each fitted elastic net model, in particular, the most regularized model [15] for GEV location and scale, and the best-fitting minimum error model [15] for GEV shape (NSE = Nash–Sutcliffe efficiency [50]; KGE = Kling–Gupta efficiency [51]; Min. = minimum; Max. = maximum; St. dev. = standard deviation; NA = not available).

| | | GEV Location | | | | | GEV Scale | | | | | GEV Shape | | | | |
|---|---|---|---|---|---|---|---|---|---|---|---|---|---|---|---|---|
| | | nfolds | | | | | | | | | | | | | | |
| | | 3 | 9 | 11 | 15 | 22 | 3 | 9 | 11 | 15 | 22 | 3 | 9 | 11 | 15 | 22 |
| NSE | Max. | 0.980 | 0.978 | 0.978 | 0.977 | 0.976 | 0.919 | 0.898 | 0.898 | 0.896 | 0.685 | −0.322 | −7.664 | −1.683 | −1.683 | NA |
| | Min. | 0.737 | 0.931 | 0.952 | 0.956 | 0.976 | −0.324 | 0.465 | 0.552 | 0.586 | 0.685 | −41,934.587 | −263.274 | −400.794 | −400.794 | NA |
| | Mean | 0.949 | 0.970 | 0.971 | 0.973 | 0.976 | 0.655 | 0.691 | 0.697 | 0.698 | 0.685 | −1181.861 | −30.333 | −35.422 | −35.615 | NA |
| | St. dev. | 0.044 | 0.008 | 0.007 | 0.005 | 0.000 | 0.251 | 0.067 | 0.057 | 0.054 | 0.000 | 6547.948 | 37.485 | 61.570 | 65.879 | NA |
| $R^2$ | Max. | 0.981 | 0.979 | 0.979 | 0.979 | 0.978 | 0.922 | 0.909 | 0.909 | 0.908 | 0.863 | 0.593 | 0.352 | 0.541 | 0.541 | NA |
| | Min. | 0.906 | 0.946 | 0.959 | 0.962 | 0.978 | 0.841 | 0.851 | 0.856 | 0.858 | 0.863 | 0.155 | 0.176 | 0.165 | 0.165 | NA |
| | Mean | 0.962 | 0.973 | 0.974 | 0.976 | 0.978 | 0.872 | 0.865 | 0.865 | 0.866 | 0.863 | 0.230 | 0.214 | 0.216 | 0.218 | NA |
| | St. dev. | 0.018 | 0.006 | 0.006 | 0.004 | 0.000 | 0.024 | 0.009 | 0.009 | 0.010 | 0.000 | 0.096 | 0.041 | 0.061 | 0.066 | NA |
| KGE | Max. | 0.978 | 0.972 | 0.972 | 0.971 | 0.969 | 0.951 | 0.924 | 0.924 | 0.922 | 0.650 | 0.189 | −1.459 | −0.275 | −0.275 | NA |
| | Min. | 0.649 | 0.901 | 0.934 | 0.938 | 0.969 | 0.050 | 0.470 | 0.533 | 0.560 | 0.650 | −200.561 | −14.500 | −18.227 | −18.227 | NA |
| | Mean | 0.930 | 0.959 | 0.961 | 0.964 | 0.969 | 0.668 | 0.662 | 0.666 | 0.667 | 0.650 | −11.682 | −3.635 | −3.761 | −3.739 | NA |
| | St. dev. | 0.061 | 0.013 | 0.011 | 0.009 | 0.000 | 0.215 | 0.075 | 0.068 | 0.066 | 0.000 | 31.650 | 1.952 | 2.743 | 2.827 | NA |

### 3.2. Modelling Spatial Dependence

The dependence modelling was limited to six streamflow gaging sites in the Current River Basin with a common period of record from 2002 to 2020. In particular, the analysis considered the six gaging stations with IDs 4–7, 9, and 10 (Figure 1, Table 2). With six sites, there were 15 potential pairs ($\binom{6}{2}$). Figure 12 is a plot of extremal coefficient estimates (estimated via the madogram [19]) as a function of Euclidean distance for those 15 possible pairs, with blue crosses for flow-connected pairs and black circles for flow-unconnected pairs.

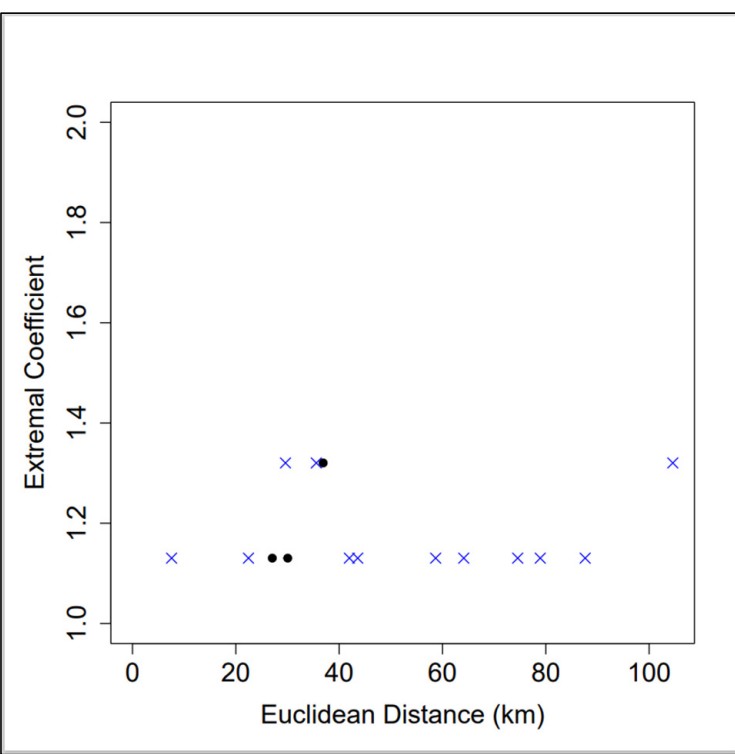

**Figure 12.** Extremal coefficients (estimated using the madogram) of all pairs of gaging stations plotted against Euclidean distance; those for flow-connected pairs are blue crosses, and those for flow-unconnected pairs are black circles.

Brown–Resnick MSP models were fit to independent multivariate events extracted from the common discharge data, transformed to unit Fréchet margins [22], for the six sites in the Current River Basin while considering different permutations of the non-Euclidean dependence model presented in Equation (4). The first model fit only considered the second component of Equation (4) ($\lambda_{Riv} = 0$), assumed isotropic data ($\beta = \frac{\pi}{2}$ and $c = 1$), and specified each gaging site's hydrologic location, $H$, to be its basin outlet rather than the center of mass of the precipitation climatology (i.e., $H(s) = s$). Hence, only the dependence parameters $\lambda_{Hydro}$ and $\alpha$ were specified as adjustable. The second fit was the same as the first but incorporated the matrix $R$ to account for geometric anisotropy. The third and fourth fits were the same as the first and second, respectively, with the exception that each gaging site's hydrologic location, $H$, was specified to be its basin centroid rather than outlet. A final fit combined both terms of Equation (4). The computed log-likelihood values associated with each model fit were 526.44, 530.09, 556.18, 574.32, and 582.75, respectively.

Using the best-fitting MSP model that applied the complete dependence model of Equation (4), Figure 13a plots model-based extremal coefficient estimates as a function of hydrologic distance.

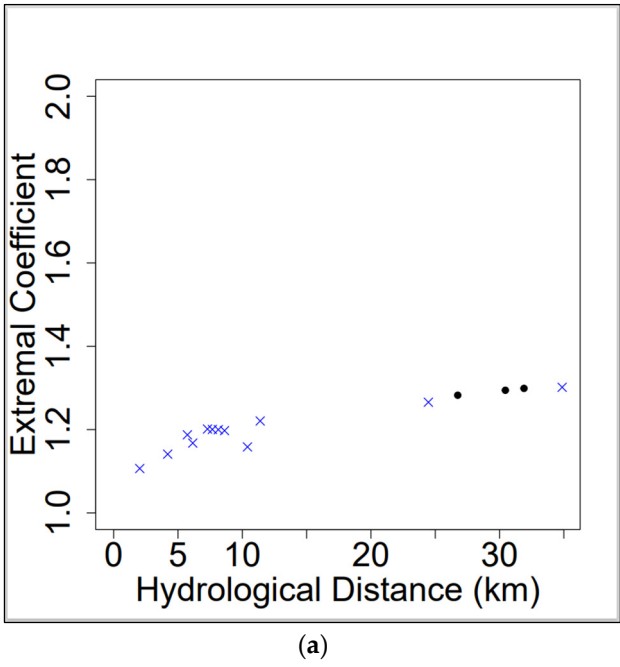 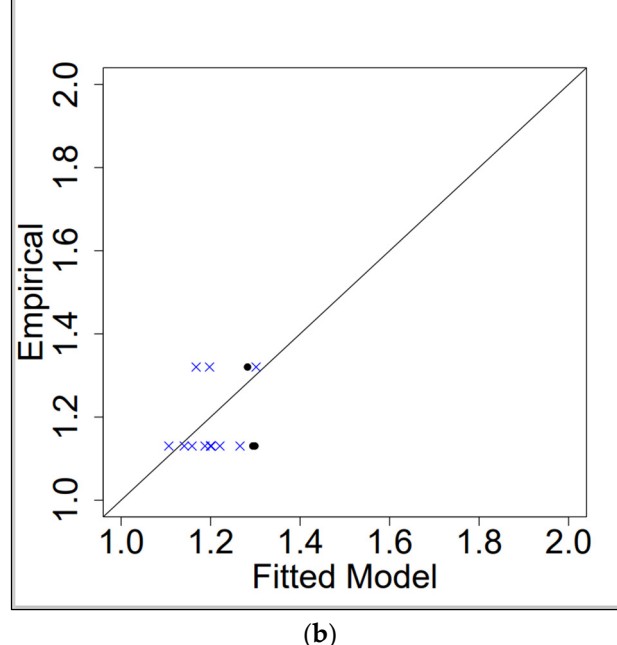

(**a**) (**b**)

**Figure 13.** Fitted MSP model-based extremal coefficient estimates obtained using Equation (4) (**a**) plotted against hydrological distance and (**b**) madogram-based estimates. Blue crosses denote flow-connected pairs and black circles signify flow-unconnected pairs.

## 4. Discussion

### 4.1. Marginal Fitting

The GEV parameter estimates listed in Table 5 are only applicable at their respective gaging site locations. The location and scale parameter estimates were subsequently used as the response variables in two independent elastic-net regression models to identify trend surface covariates for the GEV location and scale parameters. We then fit an MSP based model that used these elastic-net regression models to inform the makeup of trend surfaces for the location and scale parameters. This approach allowed us to estimate marginal distributions throughout the entire river network, $T$.

Selected covariates for the GEV location parameter included, for each gaging station's contributing drainage basin, its outlet elevation, centroid x coordinate, centroid elevation, area, average basin width [38], elevation range, and median land surface slope (Figure 9a). Covariates selected for the GEV scale parameter included area, average basin width, and median land surface slope (Figure 9b). Because of the regions' relative homogeneity, the GEV shape parameter was assumed to be constant throughout the entire study area.

The "at-site" estimates were the 66 GEV parameter values that were computed by optimizing Equation (1) without any covariates for the study's 22 streamflow gaging station sites (Table 5). Computed values for the coefficient of determination ($R^2$), Nash–Sutcliffe efficiency (NSE) [50], and Kling–Gupta efficiency (KGE) [51] equal to 0.95, 0.95, and 0.94, respectively, further summarized, in addition to the plot presented in Figure 10, a comparison of the estimates obtained from the spatial covariate model with their corresponding "at-site" estimates (Table 5) for the GEV location parameter at the 22 streamflow gaging stations. Similarly, computed values for the $R^2$, NSE, and KGE equal to 0.85, 0.84, and 0.90, respectively, further summarized the agreement for the GEV scale parameter. For the spatial covariate model, the estimated value for the GEV shape parameter was 0.0499. The computed first, second, and third quartiles and mean of the at-site estimates for the GEV shape parameter were −0.04892, 0.20571, 0.28373, and 0.09316, respectively.

The probability plots shown in Figure 11 further summarize the quality of the fit for the spatial covariate model for modelling the marginal parameters throughout the entire river network, $T$. The computed Akaike information criterion (AIC) value for the

13-parameter spatial covariate model (AIC = 24,374.22) was slightly greater than the AIC value obtained for the 66-parameter "at-site" model (AIC = 24,223.48). The spatial covariate model can be applied to estimate pointwise return levels for any location in $T$.

Tables 6 and 7 summarize the results from 1500 CV supervised elastic net optimization runs that were performed to examine the marginal trend fitting method's sensitivity for covariate selection. For each marginal distribution parameter, the most parsimonious model was always obtained using leave-one-out CV (nfolds = 22; Table 6). While increasing the number of folds provided more stability with respect to covariate selection (Table 6), the CV runs with fewer folds introduced opportunities for a greater fit (Table 7), albeit with potentially more complex models. For example, the most predictive models, defined to be the ones with the greatest NSE/KGE/$R^2$ from among the 100 CV optimization runs performed with nfolds set equal to 3, were examined for the GEV location, scale, and shape parameters. The most regularized model, defined to be at the largest value of $\lambda$ (Equation (3)) within one standard error of the minimum [15], with the greatest NSE/KGE/$R^2$ from among the 100 CV runs for the GEV location and scale parameters were of dimensions 11 and 8, respectively. The best-fitting minimum error model [15] for the GEV shape parameter was of dimension 10. By contrast, the comparable leave one out CV models were of dimensions 8, 4, and 1, for the GEV location, scale, and shape parameters, respectively (nfolds = 22; Table 6).

Spatial models were fit, using Equation (1), for these two potential covariate models of sizes 29 and 13, respectively. The AIC value associated with the fitted spatial model with 29 covariates was 25,651.5, whereas the spatial covariate model parameterized using the leave one out CV-directed elastic net results yielded a lower AIC value of 24,374.22. One possible explanation for the higher AIC value associated with the more complex spatial covariate model was the observation that the CV-directed elastic net optimization runs could not identify a predictive covariate model for the GEV shape parameter as measured by the computed and reported NSE values (Table 7). The maximum reported NSE value for the GEV shape parameter was less than zero [50]. These results for the GEV shape parameter, obtained from a comprehensive CV-directed elastic net analysis, provided support to the assumption to treat it as a constant given the physiographic and climatological homogeneity that was observed throughout the study area. It is also worth mentioning that simple synthetic numerical experiments involving the simulation of series of block maxima from GEV(0,1,0), wherein the true shape parameter is zero, with lengths equal in value to the number of seasonal maxima reported in Table 3, can be performed to demonstrate that the variability of the at-site shape parameter estimates reported in Table 5 are not necessarily inconsistent with an assumed constant shape parameter.

This study evaluated a novel covariate selection procedure within the framework of a unique MSP for modelling flood extremes on a river network [22,31]. One advantage of the marginal distribution fitting method introduced in this study is that it is applicable for alternative methods, such as Bayesian hierarchical modelling, to spatially model flood frequency [23–29].

*4.2. Modelling Spatial Dependence*

Beyond a strong dependence among all site pairs, it is difficult to identify any observable pattern with the extremal coefficient estimates plotted versus Euclidean distance in Figure 12. This contrasts with comparable plots typically obtained for extreme precipitation, SWE, temperature, and wind data [6–8].

Each of the fitted MSP model's log-likelihood values indicated improved fits when using the distance metric proposed by Asadi et al. [22], as opposed to Euclidean distance. In addition, they demonstrated that accounting for flow-connected dependence and anisotropy further improved model fit.

The curve of the data plotted in Figure 13a closely resembles what is commonly observed with dependence model summaries for MSP applications with extreme precipitation and SWE data [6–8]. The plot also shows that flow-connected sites at the same distance can

have different extremal coefficients depending on their location in $T$. Further, in general, dependence is stronger among sites that are flow-connected. Figure 13b is a plot that compares the empirical and model-based extremal coefficient estimates. While based on a limited dataset, the observed agreement is reasonable. Flood discharge exceedances can be estimated by simulating the fitted MSP that models extreme discharge dependence on the river network and subsequently transforming the simulated values using the results from the estimated marginal distributions.

## 5. Conclusions

Modelling extremes using an MSP involves two distinct steps, trend surface fitting and modelling the inter-site extremal dependence, with each step assuming independence among the extremes and fixed margins, respectively. In this study, each step was applied to discharge data from 22 streamflow gaging stations located in the Current, Little Black, Eleven Point, Spring, and Strawberry River basins in Missouri and Arkansas. The methodology utilized here was based on a unique MSP approach specifically designed for analysis of streamflow extremes. We expanded upon the novel approach by considering a larger suite of covariates and a novel automatic covariate selection procedure. The first step involved applications of the elastic-net penalty to automatically select covariate models for the marginal distribution's location and scale parameters from among a set of 16 potential covariates representing morphometric data associated with each gaging station's contributing drainage basin. The spatial covariate model required 13 parameters, whereas the "at-site" model involved 66 parameter values. While the computed AIC value for the spatial covariate model was slightly greater than the AIC value obtained for the 66-parameter "at-site" model, the spatial covariate model could be applied to estimate pointwise return levels for any location throughout the river network, whereas the "at-site" model was only applicable at the study's 22 streamflow gaging station sites (Table 5). Application of the dependence model that involved river distance and hydrologic distance rather than Euclidean distance resulted in a better fitting model of the extreme flood data, with a dependence summary that more closely resembled what is commonly observed for dependence summaries from MSP models for extreme precipitation and SWE data. Flood exceedances can be estimated throughout the entire river network using results from applications of these two steps.

There was a moderate degree of homogeneity with respect to climate, morphometry, and physiography for the five basins whose river networks were modelled in this study. In addition, the size of the discharge dataset was somewhat limited, particularly for the dependence modelling. Further related study, focusing on larger datasets and more heterogeneous systems, possibly using modified flows datasets, for example, is recommended. Adapting the likelihood for marginal fitting to combine the systematic discharge records with temporal and causal information expansion data types could be another direction to explore to potentially expand upon the capabilities of the USACE's BestFit flood frequency analysis tool.

**Supplementary Materials:** The following supporting information can be downloaded at: https://www.mdpi.com/article/10.3390/geohazards4040030/s1, Figure S1: Mean Daily Discharge Data at each of the 22 stations; Figure S2: April–November Mean Daily Discharge Data at each of the 22 stations; Figure S3: Annual maxima of the seasonal (April–November) mean daily discharge data at each of the 22 stations; Figure S4: Threshold choice plots (Location, Scale, Shape); Table S1: Covariate values.

**Author Contributions:** Conceptualization, B.S., C.H.S. and B.T.R.; methodology, B.S., C.H.S. and B.T.R.; formal analysis, B.S.; investigation, B.S., C.H.S. and B.T.R.; writing—original draft preparation, B.S.; writing—review and editing, B.S., C.H.S. and B.T.R.; funding acquisition, B.S. All authors have read and agreed to the published version of the manuscript.

**Funding:** This research was funded by the U.S. Army Corps of Engineers Mississippi River Geomorphology and Potamology Program.

**Data Availability Statement:** The data presented in this study are available on request from the corresponding author.

**Acknowledgments:** The authors thank the three reviewers for their comments which improved this article.

**Conflicts of Interest:** The authors declare no conflict of interest.

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
