# Peer review of "Marginal Distribution Fitting Method for Modelling Flood Extremes on a River Network"

_2624-795X, doi:10.3390/geohazards4040030_

Round 1

Reviewer 1 Report

Comments and Suggestions for Authors

The manuscript presents a thorough analysis of extreme flood data on river networks in Missouri and Arkansas using a max-stable process (MSP) model. The study aims to model both the marginal distributions and spatial dependence of the data, employing a non-Euclidean distance metric. The authors utilized mean daily discharge observations from 22 United States Geological Survey streamflow gaging stations and applied the elastic-net penalty to build spatially varying trend surfaces. The results indicate promising accuracy, with coefficients of determination of 0.94 and 0.92 for the GEV location and scale parameters, respectively. The study also explores the fitting of Brown-Resnick MSP models, emphasizing the importance of hydrologic distance over Euclidean distance for improved model fit. There are some comments for improvement.

1. While the computed coefficients of determination are promising, additional validation measures, such as cross-validation or model comparison with alternative approaches, could strengthen the study's robustness.

2. A sensitivity analysis on the selected covariates for marginal fitting and their impact on the model's performance could provide insights into the stability and generalizability of the results.

3. The study focuses on spatial modeling; however, considering temporal dynamics and trends in flood data could enhance the model's accuracy and applicability over different time scales.

4. The recommendation for further research with larger datasets and more heterogeneous systems is acknowledged. Future studies could explore expanding the dataset and investigating the model's performance in diverse hydrological conditions.

5. Clearly communicate and discuss the assumptions underlying the model, especially concerning the constant assumption of the GEV shape parameter throughout the study area. Sensitivity to this assumption and its potential impact on results should be addressed.

And some minor comments to address.

6. Use passive voice within the whole manuscript.

7. Rearrange Section 2. The study area should be 2.1, the dataset used should be 2.2 and the methods should be 2.3.

8. Separate the Results and Discussion section.Separating the Results and Discussion sections into distinct segments is crucial for enhancing the clarity and organization of the manuscript. By isolating these components, the reader can more effectively navigate through the study's findings and the interpretation and implications of those findings. This separation allows for a clearer distinction between objective observations and the author's analytical insights, fostering a more coherent and reader-friendly structure. In the Results section, the manuscript can succinctly present the key outcomes of the analysis, including numerical data, tables, and figures. This section serves as a repository of raw information, allowing readers to form their initial understanding of the study's outcomes without the influence of interpretation or evaluation.Conversely, the Discussion section can then delve into the nuanced interpretation of the results. Here, the authors have the opportunity to contextualize their findings within the broader scientific landscape, discuss the implications of their results, and provide insights into the significance of observed patterns or deviations. Additionally, the Discussion section is a space for addressing limitations, suggesting avenues for future research, and drawing connections between the study and existing literature. By adopting this organizational approach, the manuscript is likely to benefit from a more streamlined and reader-centric structure, ultimately enhancing the accessibility and impact of the research.

9. Please do not use refernces within the Conclusions section. The absence of references in the Conclusions section can enhance the section's readability and accessibility for a broader audience. Readers, including those less familiar with the specific field, can more easily grasp the key contributions and implications of the research without being bogged down by numerous citations.

Author Response

Dear Reviewer,

Thank you for reviewing our manuscript. Your review comments were helpful in strengthening the manuscript and we acknowledge your contribution in the Acknowledgments section of the article.

We have improved the Introduction by:

  1. moving some references that were originally placed later in the manuscript, (please see references 9-13 at line 61 in the revised manuscript),
  2. adding a new reference, 14, at line 67 of the revised manuscript,
  3. referencing other, recent, studies also directed at spatially modelling flood frequency (please see lines 87-90 and the additional references 23-29 in the revised manuscript),
  4. slightly rewording the final paragraph of the introduction to clarify/emphasize the contribution of the study.
  1. We included 2 additional summary statistics in the abstract. These are also included in the body of the revised manuscript (see lines 692-703). A cross validation directed elastic net sensitivity analysis was performed and its results were summarized in Table 6 and Table 7 with related discussion provided in the revised manuscript at lines 711-748.
  2. A cross validation directed elastic net sensitivity analysis was performed and its results were summarized in Table 6 and Table 7 with related discussion provided in the revised manuscript at lines 711-748.
  3. We agree – thank you.
  4. We agree – thank you.
  5. We addressed this comment at lines 727-743 in the revised manuscript.
  6. We addressed this comment – related revisions are spread throughout the revised manuscript.
  7. The revised manuscript addressed this comment.
  8. The revised manuscript addressed this comment.
  9. The revised manuscript addressed this comment.

Reviewer 2 Report

Comments and Suggestions for Authors

The article is titled ‘Flood Extremes on the Current, Eleven Point, Little Black, Spring, and Strawberry River Networks in Missouri and Arkansas.’ The main purpose of this article was to demonstrate that the trend matching methodology introduced by Love et al. and used by Skahill et al. it is also useful for extreme flood data on river networks. Through its use, this study allowed for the evaluation of a larger set of potential marginal modeling covariates than originally considered. The article is interesting, but sometimes difficult to read. My comments are as follows:

- the title should be changed to reflect the content of the article

- the introduction should be supplemented, adding newer references published within the last 5 years

- methods need to be better described. I propose to present the research methods in the form of a diagram to make them better understandable to the reader. The authors present data on precipitation or air temperature in individual months, but do not specify for what period of time these data are and how these averages were calculated (for how many measuring stations or only for 1)

- the characteristics of the study area should be improved. I propose to describe individual rivers, attach a map with the marked catchment area, and place meteorological and hydrological posts. The summary table should present individual parameters of all catchments - the map with the research area should be corrected. You should indicate a contour map of the USA with individual states marked, and then present a detailed map with the analyzed research area. Each map must have a legend Some data used in the article are not up-to-date, e.g. the authors refer to the structure of land use in individual catchments, using literature from 2003 or older. It is advisable to provide current values related to land use.

- results need to be improved. In this part of the text, you can add elements of discussion and refer to the advantages and disadvantages of the method used

- conclusions need to be improved. They should contain the most important findings resulting from the work; in this part we omit references to the literature

Technical notes:

There are a lot of technical errors in the article. The entire article should be adapted to the requirements of the journal. This applies, for example, to the style of tables and the list of references

- frequent quoting of selected literature, often several times in the same paragraph, e.g. Asadi et al. (18), Adamski et al. (35) Asadi et al. was quoted as many as three times in the same paragraph, similarly to items 36 and 37

- typographical errors, e.g. in lines 249, 259 and 264 there is ‘Adamaski’ instead of ‘Adamski’ - captions under figures and tables are not very precise

- some entries provide wrong units, e.g. the caption under Figure 4 should be corrected (units in mm) and it should be (units in ). In addition, you should specify the period for which these average values are

- the units should be standardized, e.g. instead of writing sq. km. (Table 2) km2 should be provided, and the same notation should be used in the text, the same applies to other units.

Author Response

Dear Reviewer,

Thank you for reviewing our manuscript. Your review comments were helpful in strengthening the manuscript and we acknowledge your contribution in the Acknowledgments section of the article.

The title of the manuscript was changed to better reflect the content of the article.

We have improved the Introduction by:

  1. moving some references that were originally placed later in the manuscript, (please see references 9-13 at line 61 in the revised manuscript),
  2. adding a new reference, 14, at line 67 of the revised manuscript,
  3. referencing other, recent, studies also directed at spatially modelling flood frequency (please see lines 87-90 and the additional references 23-29 in the revised manuscript),
  4. slightly rewording the final paragraph of the introduction to clarify/emphasize the contribution of the study.

A schematic diagram of the marginal distribution fitting method was included in the revised manuscript (please see Figure 8).

The captions for Figure 2 and Figure 4 of the revised manuscript were modified to clarify the source of the data, its related period of time, and that the averages were basin wide, for each respective system. Please also see lines 258-261 of the revised manuscript.

Please see lines 245-252 of the revised manuscript for a discussion about newly added land use calculations, using thirty-meter resolution raster datasets representative of 2001 and 2021 from the National Land Cover Database (https://www.mrlc.gov/data), for each of the respective systems.

The original Results and Discussion section was separated into separate Results and Discussion sections for improved article clarity. A cross validation directed elastic net sensitivity analysis was performed and its results were summarized in Table 6 and Table 7 with related discussion provided in the revised manuscript at lines 711-748. Advantages of the method were referenced at lines 87-90 and 744-748 of the revised manuscript.

The Conclusions were slightly modified to emphasize the main contribution and findings of the study. References were removed from the Conclusions.

Frequent quoting of selected literature was addressed throughout the revised manuscript.

Typographical errors were addressed throughout the revised manuscript.

Captions for the Tables and Figures were modified for improved precision.

Wrong units were corrected.

The time periods associated with computed average values were included in the Table and Figure captions.

Units were standardized and uniformly used throughout the manuscript.

Reviewer 3 Report

Comments and Suggestions for Authors

This study applies a trend fitting methodology of flood extremes in river basins of USA. Overall the paper is well written and results are clearly demonstrated, as well as the proposed method is useful for extreme value analysis of floods. I only note that the research gap should be more explicitly written in the Introduction and Abstract, but other than that, I have no comments for publication. Nice work.

Author Response

Dear Reviewer,

Thank you for reviewing our manuscript. Your review comments were helpful in strengthening the manuscript and we acknowledge your contribution in the Acknowledgments section of the article.

We have improved the Introduction by:

  1. moving some references that were originally placed later in the manuscript, (please see references 9-13 at line 61 in the revised manuscript),
  2. adding a new reference, 14, at line 67 of the revised manuscript,
  3. referencing other, recent, studies also directed at spatially modelling flood frequency (please see lines 87-90 and the additional references 23-29 in the revised manuscript),
  4. slightly rewording the final paragraph of the introduction to clarify/emphasize the contribution of the study.

The abstract was also slightly reworded to clarify/emphasize the contribution of the study.

Round 2

Reviewer 1 Report

Comments and Suggestions for Authors

Thank you for addressing my comments

Reviewer 2 Report

Comments and Suggestions for Authors

The authors have improved the article.